# BLOB-Q: Boosting Low Bit ViT Quantization via Global Optimization on Model Distortion

## Abstract

In this paper, we present a novel Mixed-Precision Post Training Quantization (PTQ) approach for Vision Transformers (ViTs). Different with prior works which typically optimize the output error of current layer (layer distortion), when performing quantization, our approach directly minimizes the output error of the last layer of the model (model distortion). As model distortion is highly related to accuracy, our approach can maximally maintain the accuracy even when quantized to low bit widths. We formulate the quantization of ViTs as a model distortion optimization problem, given the constraint of size. By solving the optimization problem, the optimal bit allocation across layers, i.e., the optimal bit width of each layer, can be obtained, with minimized model distortion. Directly solving the optimization problem is an NP-hard problem. We propose to adopt the second-order term of the Taylor series expansion to approximate model distortion, where an important additivity property can be derived under the approximation. Utilizing the second-order additivity property, the optimization problem can be decomposed into sub-problems and solved efficiently in an iterative manner. Specifically, we propose a dynamic programming algorithm to solve the optimization problem and efficiently find the globally optimal solution with only linear time complexity. Extensive experiments on six ViT models demonstrate the effectiveness of our approach. Results show that our approach significantly improves state-of-the-art and can further reduce the size of ViT models to 4 bits to 6 bits without hurting accuracy.

## 1 Introduction

Inspired by the success of the Transformer architecture in Natural Language Processing (NLP) Vaswani et al. (2017); Conneau et al. (2020); Devlin et al. (2019); Liu et al. (2019); Radford et al. (2018), Vision Transformer (ViT) Dosovitskiy et al. (2021); Touvron et al. (2021); Liu et al. (2021a) has become the mainstream architecture in Computer Vision and achieved state-of-the-art results in various vision tasks, from classification Krizhevsky et al. (2012), segmentation He et al. (2017) and style transfer Gatys et al. (2016); Johnson et al. (2016) of images to the synthesis of images Karras et al. (2018). However, the high accuracy of ViT models comes at the cost of high computational complexity Xu et al. (2024a); Geng et al. (2024). ViT models typically require tens of millions or even more parameters to obtain a powerful performance. For example, DeiTTouvron et al. (2021) contains about 340 millions of parameters with 12 Transformer blocks. Such a large volume of parameters and its associated memory cost make it very difficult to deploy ViT models on resource-limited mobile devices (e.g., phones, drones, watches, automatic cars, etc).

**Main Challenges**. Quantization Han et al. (2015); Zhu et al. (2017); Zhang et al. (2018); Hubara et al. (2016); Zhou et al. (2017; 2016); Rastegari et al. (2016) is an effective approach to compress deep neural networks. In this paper, we focus on Post-Training Quantization (PTQ) for ViT models. PTQ does not require retraining or fine-tuning the model after quantization and is more efficient and feasible, since retraining itself is time-consuming, and even impossible in some cases when the training dataset is not available. What's more, quantization with mixed-precision can benefit the accuracy. Because parameters in different layers react differently to quantization, instead of using equal bit width to quantize all layers, allocating different bit widths across the layers based on their sensitivity to quantitation is more reasonable and can lead to higher precision. To find the optimal bit allocation of layers, one challenge is that search space of the hyperparameters increases exponentially with the number of layers. Given $L$ layers and $B$ bit width options, the search space is $O(B^L)$, where

in a deep neural network, $L$ can be tens or hundreds of layers. Such a huge search space could make it extremely time-consuming for a heuristic search method, such as reinforcement learning Sutton & Barto (2018), to find a solution.

**Existing Problems**. Prior PTQ works for ViT models have several drawbacks. First, most existing works only consider layer distortion, i.e., the output error of current layer, when deciding the bit width and other quantization configurations of a layer. Because output error accumulates across layers, layer distortion can not reflect the real impact on the model. Small layer distortion may lead to a large output error in the last layer of the model. Using layer distortion as an indicator to make the decision of quantization configurations is not reasonable. Second, prior works typically use heuristic search methods (e.g., reinforcement learning) to find the bit allocation of layers. The heuristic search methods have high time complexity, require considerable computing resources, and may fall into a local solution in most cases. There are some other works using gradient descent methods or analytical solutions under the assumptions of the distributions of parameters. Again, these gradient descent methods can not guarantee a global-optimal solution and usually fall into a local solution. The analytical solutions require that parameters obey specific distributions (e.g., Laplacian distribution), which may not hold in practice.

**Our Contributions**. We propose a novel Mixed-Precision PTQ approach for ViTs. Our approach directly minimizes model distortion, i.e., the output error of the last layer, to maximally maintain the accuracy of the model. By solving the minimization problem, our approach can obtain optimal bit allocation of the layers with minimal model distortion. As the model distortion minimization problem is an NP-hard problem, we thus adopt the second-order term of the Taylor series expansion to approximate model distortion. We derive an important additivity property of model distortion under second-order approximation, which is that the model distortion caused by the quantization of all layers equals to the sum of all model distortion due to the quantization of individual layers. Utilizing this additivity property, the whole problem can be decomposed into sub-problems. We then develop an efficient dynamic programming algorithm to find the global-optimal solution of the problem with only linear time complexity. To the best of our knowledge, this is the first work that proposes a method to find the global-optimal solution for mixed-precision quantization. We summarize our contributions as follows:

- We propose a new Mixed-Precision PTQ approach for ViTs, which directly minimizes model distortion caused by quantization. Our approach finds the optimal bit allocation of layers with minimized model distortion and can well maintain the accuracy even when quantized to very low bit widths.
- We propose an efficient method to solve the model distortion minimization problem by adopting second-order approximation and utilizing the additivity property. An ultra-fast dynamic programming algorithm is developed to find the global-optimal solution with only linear time complexity.
- Our approach significantly outperforms state-of-the-art. For the first time, at 6 bits, we report the PTQ results on ViTs without hurting accuracy (loss < 1%). At 4 bits, our approach noticeably improves the PTQ results up to 11.49% compared with prior arts.

## 2 RELATED WORKS

**Post-Training Quantization.** Model quantization can be generally categorized into Quantization-Aware Training (QAT) and Post-Training Quantization (PTQ), where the latter PTQ performs quantization for a trained model. Recently, PTQ is more preferred as it is more lightweight without requiring to intensive training-from-scratch. Retrain-free Nagel et al. (2020a); Li et al. (2020); Zhao et al. (2019); Frantar & Alistarh (2022) approaches have become ones of desired traits of scalable PTQ for modern models. Plenty of PTQ techniques have been explored for ViTs Liu et al. (2021b); Li et al. (2023); Liu et al. (2023a); Lin et al. (2022); Di Wu (2020); Ding et al. (2022); Li & Gu (2023), mostly at 8 bits. Few others Yuan et al. (2021) explored low bit PTQ down to 4 bits with noticeable accuracy drops. We achieve 4-bits PTQ with vastly shrinked accuracy gaps, and 6-bits PTQ with full-precision level accuracies.

**Mixed-Precision Quantization.** Adapting to different sensitivity to quantization on different layers of the model, MPQ aims to allocate varying bit widths within the model (commonly in layerwise granularity). Reinforcement-Learning (RL) based approaches like HAQ Wang et al. (2019) and

AutoQ Lou et al. (2020) pioneer MPQ towards searching optimal bit allocation, but can be costly for PTQ. Recent reconstruction based method OBQ Frantar & Alistarh (2022) are compatible with mixed-precision quantization, but are limited to layerwise optimization with greedy solver.

**Model Distortion Minimization.** To make the quantization strategy aware of task loss, model-level (Loss-aware) optimization approaches like AdaRound Nagel et al. (2020a) and BRECQ Li et al. (2020) are formulated on model-level loss to layerwisely (or blockwisely) reconstruct the quantiza-

| Feature | HAQ | LAPQ | OBQ | BRECQ | HAWQ | PTQ4ViT | **BLOB-Q** |
|---|---|---|---|---|---|---|---|
| Model-level Optimization | ✓ | ✓ | ✗ | ✗ | ✓ | ✗ | ✓ |
| Global-Optimal Solver | ✗ | ✗ | ✗ | ✗ | ✗ | ✗ | ✓ |
| Mixed-Precision | ✓ | ✗ | ✗ | ✗ | ✓ | ✗ | ✓ |
| Second-order Method | ✗ | ✓ | ✓ | ✓ | ✓ | ✓ | ✓ |

Figure 1: Comparisons of different PTQ techniques.

tion rounding. LAPQ Nahshan et al. (2021) and PTQ4ViT Yuan et al. (2021) also construct model-level objectives to search for optimal quantization parameters, but PTQ4ViT ends up local solutions, while LAPQ relies on a complicated multi-stage optimization procedure. Fig. 1 illustrates a comparison with our most related works.

## 3 PRELIMINARIES AND PROBLEM STATEMENTS

Following prior works on Mixed-Precision Quantization (MPQ), *e.g.*, Wang et al. (2019); Dong et al. (2019); Liu et al. (2021b); Chen et al. (2023), we explore the MPQ problem for ViTs. Compared to Uniform-Precision Quantization (UPQ) which uses the same bit width throughout the model, MPQ works have shown better accuracy via unevenly allocating bit widths in the model. The general goal of MPQ is to solve the best quantization configuration, *a.k.a.* searching a set of bit widths $B_l^W \in \mathbb{R}^L$ and $B_l^A \in \mathbb{R}^L$ for all $L$ layer weights and activations respectively in a model.

### 3.1 LAYER DISTORTION MINIMIZATION

Previously, most PTQ approaches Nagel et al. (2020b); Li et al. (2020); Dong et al. (2019); Liu et al. (2021b); Chen et al. (2023) avoid optimizing on a model-level objective as it is NP-hard for non-linear DNNs. They generally can be summarized into the following framework, that individually solves a series of layer-level problems capturing the local impact of quantization on current layer output, under a generic compression constraint $\mathcal{R}(\widehat{\boldsymbol{W}}) \leqslant R$ that the total size $\mathcal{R}$ of the quantized model parametrized by all layer weights $\widehat{\boldsymbol{W}} = \{\widehat{\boldsymbol{W}_1}, \widehat{\boldsymbol{W}_2}, ..., \widehat{\boldsymbol{W}_L}\}$ is under a budget size $R$:

$$\arg\min_{\widehat{\boldsymbol{W}_l}} \sum_l \Gamma(\boldsymbol{O}_l, \widehat{\boldsymbol{O}}_l), \quad \text{s.t.} \quad \mathcal{R}(\widehat{\boldsymbol{W}}) \leqslant R. \tag{1}$$

Here $\boldsymbol{O}_l$ denotes the output of the layer and $\widehat{\boldsymbol{O}}_l$ denotes the output of the quantized layer. Such layer distortion minimization framework works well for previous Uniform-Precision Quantization (UPQ) works Nagel et al. (2020b); Li et al. (2020) that reconstruct the quantized weight locally (layerwisely or blockwisely) via adjusting rounding parameters in the specific form of $\arg\min_{\widehat{\boldsymbol{W}_l}} \mathbb{E}_X(\|\boldsymbol{O}_l, \widehat{\boldsymbol{O}}_l\|_F^2)$. Most recent Mixed-Precision Quantization (MPQ) attempts Dong et al. (2019); Liu et al. (2021b); Chen et al. (2023) also deduct from this layer distortion minimization to search for optimal bit allocation resulting to ranking sensitivity by *e.g.*, nuclear forms Banner et al. (2018) and Hessian terms Dong et al. (2019).

### 3.2 MODEL DISTORTION MINIMIZATION.

Despite most previous PTQ works circumvent the intimidating global optimization and resort to exploring local effect of quantization for efficient solution, in this paper, we aim to tackle the more difficult model-level optimization. Since the effect of quantization on model distortion directly links to the eventual model accuracy, theoretically it leads to a more accurate Mixed-Precision Quantization (MPQ) solution. Formally, model distortion minimization refers to the following framework:

$$\arg\min_{\widehat{\boldsymbol{W}_l}, \widehat{\boldsymbol{A}}_l} \Gamma(\boldsymbol{O}, \widehat{\boldsymbol{O}}), \quad \text{s.t.} \quad \mathcal{R}(\boldsymbol{W}, \boldsymbol{A}) \leqslant R, \tag{2}$$

where it directly optimizes the best quantization that leads to minimal model distortion in the output of the last layer $\boldsymbol{O} = \boldsymbol{O}_L$. Unlike many MPQ works which only search for weight quantization configurations, we include both weights and activations $\widehat{\boldsymbol{W}}_l, \widehat{\boldsymbol{A}}_l$ in the MPQ optimization. We define distortion $\Gamma$ as the expected L1-distance: $\Gamma(\boldsymbol{O}, \widehat{\boldsymbol{O}}) = \mathbb{E}_X(\|\boldsymbol{O} - \widehat{\boldsymbol{O}}\|)$.

The obvious challenge to tackle the above model-level optimization framework practically is the non-trivial task to find out close form relationship between perturbation on layer weights and activations of non-linear models. Reinforcement Learning (RL) based method like HAQ Wang et al. (2019) can get close-to-optimal solution for smaller CNNs, but can be extremely time-consuming to search a solution especially when model sizes scale up in modern models like ViTs. Layerwise pruning methods with a model-level objective like LAMP Lee et al. (2020) relies on greedy solver with no guarantee on global-optimum.

### 3.3 SECOND-ORDER APPROXIMATION AND ADDITIVITY PROPERTY

Recent works on model pruning Kurtic et al. (2022); Yang et al. (2023) indicate that second-order methods are essential for accurate layerwise pruning. This encourages us to explore the possibility to adopt a second-order approach for a quantization problem instead and potentially enhance the performance of Mixed-Precision Quantization. Recalling that the formulation of second-order Taylor expansion w.r.t. model output with small perturbation $\Delta \boldsymbol{W}_l$ on layer weight $\boldsymbol{W}_l$ is $\widehat{\boldsymbol{O}} = \boldsymbol{O} + \boldsymbol{J}^\top(\boldsymbol{W}_l)\Delta \boldsymbol{W}_l + \frac{1}{2}\Delta \boldsymbol{W}_l^\top \boldsymbol{H}_l^w \Delta \boldsymbol{W}_l$, we start from applying second-order Taylor expansion at the model convergence point for both weight and activation quantization[1]:

$$\Gamma_{\widehat{\boldsymbol{W}}_l}(\boldsymbol{O}, \widehat{\boldsymbol{O}}) = \mathbb{E}_X\left(\left\|\frac{1}{2}\Delta \boldsymbol{W}_l^\top \boldsymbol{H}_l^w \Delta \boldsymbol{W}_l\right\|\right), \Gamma_{\widehat{\boldsymbol{A}}_l}(\boldsymbol{O}, \widehat{\boldsymbol{O}}) = \mathbb{E}_X\left(\left\|\frac{1}{2}\Delta \boldsymbol{A}_l^\top \boldsymbol{H}_l^a \Delta \boldsymbol{A}_l\right\|\right), \quad (3)$$

where $\Gamma_{\widehat{\boldsymbol{W}}_l}(\boldsymbol{O}, \widehat{\boldsymbol{O}}), \Gamma_{\widehat{\boldsymbol{A}}_l}(\boldsymbol{O}, \widehat{\boldsymbol{O}})$ are the distortion $\Gamma(\boldsymbol{O}, \widehat{\boldsymbol{O}})$ with the quantized $\widehat{\boldsymbol{W}}_l, \widehat{\boldsymbol{A}}_l$ respectively, and $\Delta \boldsymbol{W}_l$ and $\Delta \boldsymbol{A}_l$ are the perturbation on layer weights and activations respectively by quantization (e.g. $\Delta \boldsymbol{W}_l = \boldsymbol{W}_l - \widehat{\boldsymbol{W}}_l$), $\boldsymbol{H}_l^w, \boldsymbol{H}_l^a$ are the hessian matrices on $\boldsymbol{W}_l, \boldsymbol{A}_l$ respectively w.r.t. model output. Based on second-order Taylor approximation, we discover the following property when doing arbitrary quantization:

**Property 1** (Second-Order Additivity). *The expected model distortion $\Gamma(\boldsymbol{O}, \widehat{\boldsymbol{O}})$ between the output of a pretrained model $\boldsymbol{O}$ and a quantized one $\widehat{\boldsymbol{O}}$ on dataset $X$ can be upper bounded by the sum of model distortion due to the quantization of individual layers:*

$$\Gamma(\boldsymbol{O}, \widehat{\boldsymbol{O}}) = \mathbb{E}_X(\|\boldsymbol{O} - \widehat{\boldsymbol{O}}\|_F^2) \leqslant \sum_l \Gamma_{\widehat{\boldsymbol{W}}_l}(\boldsymbol{O}, \widehat{\boldsymbol{O}}) + \sum_l \Gamma_{\widehat{\boldsymbol{A}}_l}(\boldsymbol{O}, \widehat{\boldsymbol{O}}), \quad (4)$$

We provide a mathematical derivation for the additivity property in Appendix A and empirical evidences in Appendix A.2. This property also tells us that instead of optimizing the original model distortion which is NP-hard, we can instead solve an integer programming problem on layerwise contributions to the model distortion as in below Corollary.

**Corollary 1** (Second-Order Model Distortion Optimization). *Given Eq. 20, the model distortion minimization problem in Eq. 2 can be turned into additive global objective:*

$$\underset{\widehat{\boldsymbol{W}}_l, \widehat{\boldsymbol{A}}_l}{\arg\min} \sum_l \Gamma_{\widehat{\boldsymbol{W}}_l}(\boldsymbol{O}, \widehat{\boldsymbol{O}}) + \sum_l \Gamma_{\widehat{\boldsymbol{A}}_l}(\boldsymbol{O}, \widehat{\boldsymbol{O}}), \quad \text{s.t.} \quad \mathcal{R}(\widehat{\boldsymbol{W}}, \boldsymbol{A}) \leqslant R. \quad (5)$$

Since bit-widths $B_l^W, B_l^A$ of single layer only control the quantization of single layer weights $\widehat{\boldsymbol{W}}_l$ and activations $\widehat{\boldsymbol{A}}_l$, they also only affect $\Gamma_{\widehat{\boldsymbol{W}}_l}$ and $\Gamma_{\widehat{\boldsymbol{A}}_l}$ of current layer. Furthermore, as the size constraint $\mathcal{R}$ can be instantiated as the linear function of bit-widths *i.e.*, $\mathcal{R}(\boldsymbol{W}, \boldsymbol{A}) = \sum_l B_l^W\|\boldsymbol{W}_l\|_0 + B_l^A\|\boldsymbol{A}_l\|_0$, The model distortion minimization problem in Eq. 5 is equivalent to a integer programming problem:

$$\underset{B_l^W, B_l^A}{\arg\min} \sum_l \Gamma_{\widehat{\boldsymbol{W}}_l}(\boldsymbol{O}, \widehat{\boldsymbol{O}}) + \sum_l \Gamma_{\widehat{\boldsymbol{A}}_l}(\boldsymbol{O}, \widehat{\boldsymbol{O}}), \quad \text{s.t.} \quad \sum_l B_l^W\|\boldsymbol{W}_l\|_0 + B_l^A\|\boldsymbol{A}_l\|_0 \leqslant R, \quad (6)$$

with only the concrete values of $\Gamma_{\widehat{\boldsymbol{W}}_l}$ and $\Gamma_{\widehat{\boldsymbol{A}}_l}$ under different bit widths needed to be obtained via calibration, detailed in Section. 4. Supported by the additivity property in Eq. 20, we can guarantee to attain the global-optimal of the Model-level objective in Eq. 2 by solving the integer programming problem. We eventually devise a non-greedy solver for problem Eq. 6 detailed in Section. 5.

---

[1]Prior works indicate first-order term can be regarded vanished on converged model

## 4 Calibration for Second-Order Model Distortion

There are still remaining major challenges to leverage Property 1 for quantization practically for empirical ViTs. For $\boldsymbol{H}^w$, some existing second-order *pruning* methods Kurtic et al. (2022); Yang et al. (2023) pointed out some on possible ways for us. For $\boldsymbol{H}^a$, some second-order PTQ methods like BRECQ Li et al. (2020) involve Hessian on activations $\boldsymbol{H}^a$ but are designed for guiding weight quantization. Moreover, there was no attempt to seamlessly weld different Hessian approximation schemes (e.g. gradient Yang et al. (2023), empirical Fisher Kurtic et al. (2022), and diagonal gradient product Li et al. (2020)) for unified weight and activation MPQ search.

In this work, we unify the second-order approximation for Mixed-Precision Quantization (MPQ) search of both weights and activations. We start by selecting empirical Fisher Kurtic et al. (2022) scheme for approximating Hessian for both weight $\boldsymbol{H}^w$ and activation $\boldsymbol{H}^a$, to avoid potential mismatch of using different schemes in the unified allocation problem in Eq. 20:

**Definition 1** (Empirical Fisher Kurtic et al. (2022)). *The Hessian $\boldsymbol{H}^w \in \mathbb{R}^{d \times d}$ of weight $\boldsymbol{W} \in \mathbb{R}^{d_{row} \times d_{col}}$ ($d = d_{row}d_{col}$) can be estimated on calibration dataset $X_{cal}$ of $N$ samples given the Jacobian matrix $\boldsymbol{J}_n(\boldsymbol{W})$ of the weight matrix on $n$-th sample as $\widetilde{\boldsymbol{H}}^w = \kappa \mathbf{I}_d + \frac{1}{N}\sum_n^N \boldsymbol{J}_n(\boldsymbol{W})\boldsymbol{J}_n^\top(\boldsymbol{W})$. $\kappa$ is predefined hyperparameter.*

Empirical Fisher was originally used to approximate Hessian of layer weights in pruning. We attempt to similarly approximate Hessian of layer activations in this work as $\widetilde{\boldsymbol{H}}^a = \kappa \mathbf{I}_d + \frac{1}{N}\sum_n^N \boldsymbol{J}_n(\boldsymbol{A})\boldsymbol{J}_n^\top(\boldsymbol{A})$. However, even with such Hessian estimation, it is obviously still too expensive for processing real models to store large chunks of Hessian matrices for all layers. To solve global objective in Eq. 5, we need to sample all values of $\Gamma_{\widehat{\boldsymbol{W}}_l}$ and $\Gamma_{\widehat{\boldsymbol{A}}_l}$ of $L$ layers under all possible quantized $\widehat{\boldsymbol{W}}_l$ and $\widehat{\boldsymbol{A}}_l$ during calibration. Therefore, we need $\Theta(\sum_l d_l^2)$ memory space to accommodate all layers Hessian matrices alone, which is near quadratic to the model size. In fact, there are still plenty of rooms for overhead and memory reductions, as we show that the expensive time and spatial costs can be significantly reduced in the followings.

**Second-Order Approximation for Weights.** For approximating distortion caused by weight quantization $\Gamma_{\widehat{\boldsymbol{W}}_l}$, to obtain Hessian matrix based on empirical Fisher, we first consider using the approximated Hessian (**Definition** 1) to convert the second-order term of Eq. 20 into

$$\frac{1}{2}\Delta \boldsymbol{W}^\top \left( \kappa \mathbf{I}_d + \frac{1}{N}\sum_n^N \boldsymbol{J}_n(\boldsymbol{W})\boldsymbol{J}_n^\top(\boldsymbol{W}) \right) \Delta \boldsymbol{W} = \frac{\kappa}{2}\|\Delta W\|_2^2 + \frac{1}{2N}\sum_n \|\boldsymbol{J}_n^\top(\boldsymbol{W})\Delta \boldsymbol{W}\|_2^2, \quad (7)$$

where we notice it no longer needs to store the full Hessian and instead mainly the inner product between $\boldsymbol{J}_n(\boldsymbol{W}), \Delta \boldsymbol{W} \in \mathbb{R}^d$ which is a scalar value. The asymptotic time and spatial complexity of this form is then $\Theta((2N+2)d+N)$ and $\Theta(2d+N)$ respectively, which only has linear complexity w.r.t. the parameter size $d$. This form is still space for further improvement to parallelize the $\sum_n \|\boldsymbol{J}_n^\top(\boldsymbol{W})\Delta \boldsymbol{W}\|_2^2$ for all $N$ samples to trade for more time efficiency with parallel computing on GPUs. This can be done by concatenating N samples of Jacobians into $\boldsymbol{J}_N(\boldsymbol{W}) \in \mathbb{R}^{N \times d}$ and obtain a $N$-dimensional variable $\mathbf{P}^w = \boldsymbol{J}_N^\top(\boldsymbol{W})\Delta \boldsymbol{W} \in \mathbb{R}^N$. Then the time cost can be slightly reduced to $\Theta((N+2)d+2N)$ by calculating $\frac{1}{2N}\text{trace}(\mathbf{P}^w \mathbf{P}^{w\top})$ instead, with slightly more space $\Theta(2d+N^2)$ to store the intermediate vector $\mathbf{P}^w$. Finally, since the above empirical Fisher estimation already encapsulates an averaging process on calibration data, everything inside expectation of $\Gamma_{\widehat{\boldsymbol{W}}_l}$ (see Eq. 20) is already invariant to the data samples. Therefore the final expectation calculation step is unnecessary for weight quantization. Hence, the final second-order model distortion for weight quantization can be summarized as:

$$\Gamma_{\widehat{\boldsymbol{W}}_l}(\boldsymbol{O}, \widehat{\boldsymbol{O}}) \approx \frac{\kappa}{2}\|\Delta \boldsymbol{W}_l\|_2^2 + \frac{1}{2N}\text{trace}(\mathbf{P}_l^w \mathbf{P}_l^{w\top}). \quad (8)$$

Empirically, we observe the overhead of second-order calibration ranges from merely around 3 to 18 minutes between the smallest and the largest ViTs (ViT-S/224 and ViT-B/384) on a 4-L40 GPU server. We will give detailed efficiency analysis in the experimental section.

**Second-Order Approximation for Activations.** To scalably approximate the second-order model distortion for activation $\Gamma_{\widehat{\boldsymbol{A}}_l}$, most ideas from above $\Gamma_{\widehat{\boldsymbol{W}}_l}$ work with an exception. Since the activation map changes for different data samples, we need to keep $N$ copies of activations $\boldsymbol{A}^N =$

$\{\boldsymbol{A}^{(1)}, \boldsymbol{A}^{(2)}, ..., \boldsymbol{A}^{(N)}\}$, which raises another question how to similarly get $\boldsymbol{P}^a$ from $\boldsymbol{A}^N \in \mathbb{R}^{N \times dT}$ and $\boldsymbol{J}_N \in \mathbb{R}^{N \times dT}$ ($T$ is the number of visual tokens in ViTs) recalling from $\boldsymbol{P}^w$ above. Here we calculate it with a slight variation as $\boldsymbol{P}^a = \text{diag}(\boldsymbol{J}_N(\boldsymbol{A}^N)\Delta\boldsymbol{A}^{N\top}) \in \mathbb{R}^N$. With other parts similar to the case of weight quantization above, this gives the following formulation:

$$\Gamma_{\widehat{\boldsymbol{A}}_l}(\boldsymbol{O}, \widehat{\boldsymbol{O}}) \approx \frac{\kappa}{2}\|\Delta\boldsymbol{A}_l\|_2^2 + \frac{1}{2N}\text{trace}(\boldsymbol{P}_l^a \boldsymbol{P}_l^{a\top}). \tag{9}$$

The time and space complexity would be $\Theta((N^2+N)\|s_i\|_0 + N)$ and $\Theta(2N\|s_i\|_0 + N^2)$ respectively, which are still only linear complexity. We have a more detailed analysis in supplemental materials.

**Efficiency of Second-Order Calibration.** After the above optimization, the calibration process is now streamlined to a one-time forward+backward pass on calibration set to obtain necessary gradient information $\boldsymbol{J}_N(\boldsymbol{W}), \boldsymbol{J}_N(\boldsymbol{A})$, followed by applying Eq. 8 and Eq. 9 individually to rapidly approximate global output

Table 1: Time and spatial complexity of the second-order calibration for whole model. $D_W$ and $D_A$ denote the total parameter size and total feature map size for a model respectively.

|  | Weight | Activation |
|---|---|---|
| Time | $\Theta(Nl + (N+2)D_W)$ | $\Theta((N^2+N)D_A)$ |
| Space | $\Theta(2\max_l \|\boldsymbol{W}_l\|_0 + N^2 l)$ | $\Theta(2N\max_l \|\boldsymbol{A}_l\|_0 + N^2 l)$ |

distortions $\Gamma_{\boldsymbol{W}_l}, \Gamma_{\boldsymbol{A}_l}$ under arbitrary quantizing weights and activations. In practice, one may iterate through a finite set of different quantization configurations to sample $\widehat{\boldsymbol{W}}, \widehat{\boldsymbol{A}}$, *e.g.* a set of possible bit-widths and quantization step sizes, etc. We summarize the estimated end-to-end time and space complexity analysis in Tab. 1.

The total calibration time costs scale linearly with the sizes of parameters and activation states, and the maximum memory required is capped at the layer with largest weight or activation footprints ($\|\boldsymbol{W}_l\|_0$ and $\|\boldsymbol{A}_l\|_0$). In Tab. 1 we omit the cost for quantization itself. This calibration scheme is also compatible with blockwise MPQ (assign bit-width on a block of weight/activation within layer). Compared to vanilla layerwise MPQ, blockwise MPQ does not increase much time overhead and has less memory requirement as one can deduct from Tab. 1, as it can be seen as changing the problem space with now total number of weight/activation blocks beccoming the number of layers and each with less parameter sizes. We explain more blockwise MPQ settings in experiment section.

## 5 GLOBAL OPTIMAL SOLVER VIA DYNAMIC PROGRAMMING

We adopt Dynamic Programming (DP) algorithm to recursively solve the integer problem by decomposing the whole problem into multiple sub-problems and then processing each sub-problem separately. Defining the state function required by DP as $S(l, r) \in \mathbb{R}$ is a function of discrete values layer $l \in [1, L]$ and remaining model total bits $r \in [1, R]$, this indicates the the minimal achievable model distortion at current intermediate state. Then the DP algorithm can be completed by the following state transferring function to recursively solve the original problem at state $S(L, R)$:

$$S(l, r) = \min_{1 \leqslant b_1 \leqslant B}\{\Gamma_{q(\boldsymbol{Z}_l, b_1)}(\boldsymbol{O}, \widehat{\boldsymbol{O}}) + S(l-1, r - b_1 d_l)\}, \tag{10}$$

where $\boldsymbol{Z}_l$ represents the either one between $\boldsymbol{W}_l$ and $\boldsymbol{A}_l$ and we treat states for weights and activates the same as they both contribute to the total bit budget $R$. $q(\boldsymbol{Z}_l, b)$ is the optimal $b$-bit quantizer for input $\boldsymbol{Z}_l$, $B$ is the maximum allowed bit-width for each layer (we set to $B = 10$). We set the boundary condition of $S(0, :) = 0$. As we populate through the above states $S(l, r)$, we use another function $G(l, r) \in \mathbb{Z}_+$ to record all intermediate selections of $b_1$ made:

$$G(l, r) = \arg\min_{1 \leqslant b_1 \leqslant B}\{\Gamma_{q(\boldsymbol{Z}_l, b_1)}(\boldsymbol{O}, \widehat{\boldsymbol{O}}) + S(l-1, r - b_1 d_l)\}. \tag{11}$$

Using $G(l, r)$, we can iterate through layers to find the optimal bit-width $B_l$ at layer $l$.

The theoretical time complexity of the DP-based solver is $O(2LRB^2)$, as we have $2L \times R$ different states since we consider two variable bit-widths *i.e.* weight and activation for each layer. It shows the algorithm scales linearly with the model size. Moreover, the solution found by dynamic programming is the globally optimal solution. Taking advantages of modern parallel compute like GPU, the DP-solver can be further accelerated towards practical usages. We provide more details of our

---

**Algorithm 1** BLOB-Q: Model-level Global-optimal Mixed-precision Quantization Algorithm.

---

**Require:** Calibration dataset $X_{cal}$ with $N$ samples, pretrained model with $L$ layers, maxmimum allowed bit-width $B$, target total model size $R$.
**Ensure:** Optimal bit-widths for layer Weights and activations $B_l^W, B_l^A \in \mathbb{R}, 1 \leqslant l \leqslant L$.
 1: A forward+backward pass on $X_{cal}$ to get $\boldsymbol{A}_N^{(l)}, J_N(\boldsymbol{W}_l), J_N(\boldsymbol{A}_l)$      ▷ Start calibration.
 2: **for each** layer $l \in [1, L]$ **do**
 3:      **for each** bit $b \in [1, B]$ **do**
 4:          $\widehat{\boldsymbol{W}}_l = q(\boldsymbol{W}_l, b), \ \widehat{\boldsymbol{A}}_l = q(\boldsymbol{A}_l, b)$      ▷ Quantize weights and activations to $b$-bit.
 5:          $\Delta\boldsymbol{W}_l = \boldsymbol{W}_l - \widehat{\boldsymbol{W}}_l, \ \Delta\boldsymbol{A}_l = \boldsymbol{A}_l - \widehat{\boldsymbol{A}}_l$      ▷ Quantization errors.
 6:          $\boldsymbol{P}_l^w = \mathbf{J}_N^\top(\boldsymbol{W}_l)\Delta\boldsymbol{W}_l, \ \boldsymbol{P}_l^a = \mathrm{diag}(\boldsymbol{J}_N(\boldsymbol{A}_l^N)\Delta\boldsymbol{A}_l^{N\top})$
 7:          $\Gamma_{q(\boldsymbol{W}_l, b)}(\boldsymbol{O}, \widehat{\boldsymbol{O}}) = \frac{\kappa}{2}\|\Delta\boldsymbol{W}_l\|_2^2 + \frac{1}{2N}\mathrm{trace}(\mathbf{P}_l^w \mathbf{P}_l^{w\top})$. ▷ Model Distortion w.r.t. W-quant. (Eq. 8)
 8:          $\Gamma_{q(\boldsymbol{A}_l, b)}(\boldsymbol{O}, \widehat{\boldsymbol{O}}) = \frac{\kappa}{2}\|\Delta\boldsymbol{A}_l\|_2^2 + \frac{1}{2N}\mathrm{trace}(\boldsymbol{P}_l^a \boldsymbol{P}_l^{a\top})$.    ▷ Model Distortion w.r.t. A-quant. (Eq. 9)
 9:      **end for each**
10: **end for each**
11: Create matrices S, G $\in \mathbb{R}^{2L+1, R}$      ▷ Start DP-solver (Sec.5).
12: Initialize S$[1, r] \leftarrow 0, \ \forall r \in [1, R]$ if $l = 0$ else $\infty$.
13: **for each** layer $l \in [1, 2L]$ **do**      ▷ (have $2L$ for both weights and activations)
14:      **for each** size $r \in [1, R]$ **do**
15:          Update S$[1, r]$ using Eq. 10, Update G$[1, r]$ using Eq. 11      ▷ Recursively populate states.
16:      **end for each**
17: **end for each**
18: $B_L^W = $ G$[2L, R]$      ▷ Last layer weight bit-width.
19: **for each** layer $l$ reversely from $L$ to 1 **do**      ▷ Recursively obtain layerwise bit-widths.
20:      $B_l^A = $ G$[2l, \ R - B_l^W\|\boldsymbol{W}_l\|_0]$      ▷ optimal activation bit-width.
21:      $R \leftarrow R - B_l^W\|\boldsymbol{W}_l\|_0$
22:      $B_l^W = $ G$[2l-1, \ R - B_l^A\|\boldsymbol{A}_l\|_0]$      ▷ layerwise weight bit-width.
23:      $R \leftarrow R - B_l^A\|\boldsymbol{A}_l\|_0$
24: **end for each**

---

implemented DP-solver in the supplementary materials, incorporating strategies like vectorized updating and Run-length coding, with source code attached. Empirically, we find that DP-solver only requires to perform in few seconds as we show in Tab. 5. We provide the end-to-end pseudocode of our proposed BLOB-Q algorithm in Algo. 1.

To this end, we proposed a unified second-order approximation scheme for model output distortion caused by the quantization of both weights and activations. We started by discovering an important property on ViT models under second-order taylor approximation, which allows us to transform the NP-hard Global distortion optimization problem into subproblems. This allows us to solve the model distortion minimization problem in a global-optimal and non-greedy manner via Dynamic Programming. Finally, utilizing Hessian matrix information, we developed an ultra-fast algorithm to calculate the second-order distortion during post-training calibration efficiently considering the practical settings.

## 6   EXPERIMENTS

**Implementation Details.** We randomly select 64 samples from the ImageNet training set as the calibration dataset without any augmentations, and evaluate the quantized ViTs on full validation set. We conduct all experiments on Nivida 4-L40 with 48GB of VRAM on each GPU. For bias correction, we adopt the strategies same as Liu et al. (2021b) to rectify the layerwise output distribution. For finetuning, we use the same optimizer and criterion as original ViT strategy, with learning rate `lr` $= 10^{-6}$ with batchsize of 128 for larger ViT-B/384 and DeiT-B/384 and 256 for the rest. We quantize all the Conv2D and Linear layers in ViTs, including the first and last layer and excluding LayerNorm and Embedding layers. For quantizer, we choose vanilla symmetric uniform quantizer for both weights and activations. We perform blockwise MPQ in the experiments as earlier introduced in Section 4, *i.e.* allocating a group of channels in the layer weight/activation one time, where we set the blocksize to 64-channels for larger ViT-B/384 and DeiT-B/384 and 32-channel for the rest. We provide more details in supplementary materials.

**Main Results.** We compare our approach with prior Post-Training Quantization (PTQ) methods for ViTs at six ViT models (ViT-S, ViT-B, ViT-B/384, DeiT-S, DeiT-B, and DeiT-B/384) on the ImageNet validation dataset, including PTQ Liu et al. (2021b), PTQ4ViT Yuan et al. (2022), Percentile Li et al. (2019), EasyQuant Wu et al. (2020), APQ-ViT Ding et al. (2022), NoisyQuant Liu et al. (2023b), I-ViT Li & Gu (2023), FQ-ViT Lin et al. (2021), PMQ Xiao et al. (2023), P²-ViT Shi et al. (2024) and PTMQ Xu et al. (2024b). Tab. 2 illustrates the results. Our approach consistently outperforms the baseline methods in most of the cases, especially at low bit widths. Spefically, at 4 bits, our approach outperforms other methods by 1.72% and 11.49% on ViT-S and ViT-B, respectively. On DeiT-S and DeiT-B, at 4 bits, our approach outperforms others by 6.15% and 3.54%, respectively. It is worth mentioning that our approach can quantize the ViT models to 6 bits without hurting accuracy (loss < 1%). We notice that most method can obtain very well accuracy at 8 bits. This is because that 8 bits are long enough to maintain the precision.

Table 2: Results on the ImageNet validation dataset when ViT mdoels are quantized to 4 bits, 6 bits, and 8 bits. Top-1 classification accuracy is evaluated. MP denotes quantization with Mixed-Precision. For a fair comparison, we report the results with the same average bit widths. For example, 4 MP means that we quantize the ViT Models into 4 bits on average across the layers. The best result of each group is highlighted with bold.

| Method | Size (Bit) | ViT-S | ViT-B | ViT-B/384 | DeiT-S | DeiT-B | DeiT-B/384 |
|---|---|---|---|---|---|---|---|
| Full Precision | 32 | 81.39% | 84.54% | 86.05% | 79.87% | 81.85% | 83.12% |
| PTQ4ViT Yuan et al. (2022) | 4 | 42.57% | 30.69% | - | 34.08% | 64.39% | - |
| APQ-ViT Ding et al. (2022) | 4 | 47.95% | 41.41% | - | 43.55% | 67.48% | - |
| RepQ-ViT Li et al. (2023) | 4 | 65.60% | 68.48% | - | 69.03% | 75.61% | - |
| P²-ViT Shi et al. (2024) | 4 MP | 64.24% | 79.93% | - | 75.26% | 79.37% | - |
| BLOB-Q | 4 MP | **67.32%** | **79.97%** | 71.20% | **77.42%** | **80.30%** | 80.17% |
| PTQ Liu et al. (2021b) | 6 MP | 70.24% | 75.26% | 46.88% | 75.10% | 77.47% | 68.44% |
| PTQ4ViT Yuan et al. (2022) | 6 | 78.63% | 81.65% | 83.34% | 76.28% | 80.25% | 81.55% |
| Percentile Li et al. (2019) | 6 MP | 67.74% | 77.63% | 77.60% | 70.49% | 73.99% | 78.24% |
| EasyQuant Wu et al. (2020) | 6 | 75.13% | 81.42% | 82.02% | 73.26% | 75.86% | 81.26% |
| APQ-ViT Ding et al. (2022) | 6 | 79.10% | 82.21% | - | 77.76% | 80.42% | - |
| RepQ-ViT Li et al. (2023) | 6 | 80.43% | 83.62% | - | 77.76% | 80.42% | - |
| NoisyQuant Liu et al. (2023b) | 6 | 78.65% | 82.32% | 83.22% | 77.43% | 80.70% | 81.65% |
| PMQ Xiao et al. (2023) | 6 MP | - | 73.33% | - | 76.68% | 79.64% | - |
| P²-ViT Shi et al. (2024) | 6 MP | 32.05% | 82.1% | - | 77.56% | 80.59% | - |
| PTMQ Xu et al. (2024b) | 6 MP | 76.09% | 77.7% | - | 78.74% | 80.81% | - |
| BLOB-Q | 6 MP | **80.73%** | **84.23%** | **85.20%** | **79.53%** | **81.85%** | **82.72%** |
| PTQ Liu et al. (2021b) | 8 MP | 80.46% | 76.98% | 85.35% | - | 75.94% | - |
| PTQ4ViT Yuan et al. (2022) | 8 | 81.00% | 84.25% | 85.82% | 79.47% | 81.48% | 82.97% |
| Percentile Li et al. (2019) | 8 MP | 78.77% | 80.12% | 82.53% | 73.98% | 75.21% | 80.02% |
| EasyQuant Wu et al. (2020) | 8 | 80.75% | 83.89% | 85.53% | 76.59% | 79.36% | 82.10% |
| APQ-ViT Ding et al. (2022) | 8 | 81.25% | 84.26% | - | 79.78% | 81.72% | - |
| NoisyQuant Liu et al. (2023b) | 8 | 81.15% | 84.22% | 85.86% | 79.51% | 81.45% | 82.49% |
| I-ViT Li & Gu (2023) | 8 | 81.27% | 84.76% | - | 80.12% | 81.74% | - |
| FQ-ViT Lin et al. (2021) | 8 | - | 83.31% | - | 79.17% | 81.20% | - |
| P²-ViT Shi et al. (2024) | 8 MP | 68.14% | 83.00% | - | 78.41% | 80.93% | - |
| BLOB-Q | 8 MP | **81.54%** | **84.89%** | **85.93%** | **81.25%** | **81.75%** | **82.85%** |

**Ablation Studies.** Both model distortion minimization and global-optimal solution contribute to the results. We conduct an ablation study to evaluate the effectiveness of each of them, which is shown in Tab. 3. In the ablation study, we evaluate the results of three methods which are (a) minimize layer distortion with a local solution, (b) minimize model distortion with a local solution, and (c) minimize model distortion with a global-optimal solution. As we can see in Tab. 3, both of model distortion minimization and global-optimal solution can make a noticeable impact on the accuracy. For example, at 4 bits, with model distortion minimization, the accuracy is improved from 0.11 to 20.52 on ViT-S. With a global-optimal solution, the accuracy is improved from 20.25 to 61.45. Similarly noticeable results can be also observed on other models and bit widths.

**Impact of the Calibration Set.** We evaluate the impact of the calibration set on our approach. Tab. 4 illustrates the results of our approach with 32 and 64 calibration images. First, our approach only requires a very small calibration set to obtain the statistical data of model distortion. We notice that increasing the calibration size from 32 images to 64 images, our approach can obtain slightly higher accuracy. With more calibration images (e.g., 128), the accuracy can not be improved any more but

Table 3: Ablation studies of our approach. The effectiveness of model distortion minimization and global-optimal solution are evaluated. We use a greedy algorithm to find a local solution in this table. For layer distortion, we use the output error of current layer in the learning objective. The supplementary materials provide more details about the local solution.

| Comparison | Size | ViT-S | ViT-B | ViT-B/384 | DeiT-S | DeiT-B | DeiT-B/384 |
|---|---|---|---|---|---|---|---|
| Layer Distortion + Local | 4 Bits | 0.11% | 0.10% | 0.09% | 0.10% | 0.12% | 0.12% |
| Model Distortion + Local | 4 Bits | 20.52% | 1.25% | 0.60% | 0.16% | 16.51% | 49.92% |
| Model Distortion + Global | 4 Bits | **61.45%** | **77.31%** | **51.69%** | **75.14%** | **79.21%** | **79.46%** |
| Layer Distortion + Local | 6 Bits | 0.10% | 0.12% | 0.11% | 0.17% | 14.01% | 50.80% |
| Model Distortion + Local | 6 Bits | 62.87% | 15.27% | 11.04% | 40.25% | 44.08% | 74.70% |
| Model Distortion + Global | 6 Bits | **78.88%** | **83.47%** | **84.71%** | **79.21%** | **81.58%** | **82.72%** |
| Layer Distortion + Local | 8 Bits | 39.81% | 1.25% | 14.43% | 0.12% | 45.81% | 76.08% |
| Model Distortion + Local | 8 Bits | 67.78% | 52.34% | 61.23% | 74.88% | 80.81% | 81.12% |
| Model Distortion + Global | 8 Bits | **81.37%** | **84.78%** | **85.69%** | **79.64%** | **81.76%** | **82.99%** |

Table 4: Impact of calibration size on the results. We randomly select 32 or 64 images from the ImageNet training dataset as the calibration set.

| Bit Width | 4 Bits | | 6 Bits | | 8 Bits | |
|---|---|---|---|---|---|---|
| Calibration Size | 32 Images | 64 Images | 32 Images | 64 Images | 32 Images | 64 Images |
| ViT-S | 51.04% | 59.71% | 81.90% | 79.98% | 83.26% | 81.30% |
| ViT-B | 78.82% | 74.51% | 84.10% | 83.58% | 84.46% | 83.95% |
| ViT-B/384 | 66.01% | 52.09% | 84.42% | 85.18% | 85.35% | 86.01% |
| DeiT-S | 68.75% | 72.80% | 78.12% | 78.97% | 81.25% | 79.51% |
| DeiT-B | 78.12% | 78.96% | 81.25% | 81.81% | 81.25% | 81.78% |
| DeiT-B/384 | 78.59% | 79.06% | 82.25% | 82.72% | 82.59% | 82.56% |

the computation complexity can be significantly increased. We thus use 64 calibration images in all of our experiments.

**Executive Time.** We also evaluate the execution time of the optimization method and compare it with other baselines, including PTQ4ViT Yuan et al. (2022), REPQ-ViT Li et al. (2023), EasyQuant Wu et al. (2020), and NoisyQuant Liu et al. (2023b). Our method is much faster than others by orders of magnitude. As shown in Tab. 5, our method is $72\times$ faster than PTQ4ViT Yuan et al. (2022), $85\times$ faster than REPQ-ViT Li et al. (2023), $146\times$ faster than EasyQuant Wu et al. (2020), and $543\times$ faster than NoisyQuant Liu et al. (2023b). Different with prior works which need to search the solution from a huge space with exponential complexity, our method utilizes the additivity property and adopts dynamic programming find the global-optimal solution. As a result, our optimization algorithm has only linear time complexity and is much faster than others. We anaylzed the time and memory efficiency in more detail in Appendix D.4.

Table 5: Evaluation of the executive time of optimization. We test the time of our dynamic programming algorithm to find the global-optimal bit allocation. For other baseline methods, we run their code on the same hardware (Nvidia L40) to evaluate the time.

| Method | Executive Time of Optimization Method (s) | | | | | | | | | | | |
|---|---|---|---|---|---|---|---|---|---|---|---|---|
| | ViT-S | | | ViT-B | | | DeiT-S | | | DeiT-B | | |
| | 4 Bits | 6 Bits | 8 Bits | 4 Bits | 6 Bits | 8 Bits | 4 Bits | 6 Bits | 8 Bits | 4 Bits | 6 Bits | 8 Bits |
| PTQ4ViT Yuan et al. (2022) | 103.8 | 101.4 | 101.4 | 193.2 | 182.4 | 196.2 | 100.6 | 100.2 | 100.8 | 196.8 | 198.0 | 207.6 |
| REPQ-ViT Li et al. (2023) | 123.4 | 126.1 | 118.8 | 247.0 | 252.8 | 114.2 | 119.0 | 119.1 | 244.6 | 251.9 | 251.6 |
| EasyQuant Wu et al. (2020) | 211.1 | 211.5 | 211.1 | 463.0 | 463.6 | 464.0 | 214.3 | 212.4 | 211.2 | 463.0 | 463.6 | 463.7 |
| NoisyQuant Liu et al. (2023b) | 781.0 | 780.8 | 784.5 | 1844.5 | 1850.4 | 1851.5 | 787.6 | 777.7 | 778.3 | 1845.2 | 1848.6 | 1848.4 |
| BLOB-Q | **1.438** | **1.209** | **1.298** | **1.749** | **2.291** | **2.236** | **1.408** | **1.329** | **1.212** | **1.814** | **2.121** | **2.276** |

# 7    CONCLUSION

In conclusion, we novelly introduced BLOB-Q, a global-optimal model distortion minimized Mixed-precision Quantization method. Our approach builds on a key property observed in ViT models under second-order Taylor approximation, enabling the transformation of the NP-hard global distortion optimization problem into manageable subproblems. We further proposed a non-greedy and

lightweight DP-solver to solve MPQ bit-allocations that finds globally optimal solutions. Furthermore, we proposed a unified hessian approximation scheme, accompanied by an ultra-fast algorithm with minimal memory footprint, ensuring scalability of the method. Experiments show that BLOB-Q significantly boost PTQ performance on ViT models, and successfully bridge the performance gap on as low as 4-bit even without retraining.

## REPRODUCIBILITY STATEMENT

We ensure to provide all necessary details to reproduce the experimental results shown. In the Supplementary Materials, we provide the source code, containing:

- Experimentation scripts for second-order calibration and DP-solver to 4/6/8-bits.
- Code for bias-correction and retraining for auxiliary results.
- A README file providing sample commands and information on how to run all scripts.

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

## A  PROOF OF PROPERTY 1 (SECOND-ORDER ADDITIVITY)

We first provide the derivation of a strict Second-Order Additivity (Property 1) that we introduced in the main text. In the following section, we also provide a more general form of Second-order Additivity (Property 1.1) in the form of upper bound which requires less assumptions to proof. Our mixed-precision bit-allocation solution can optimize both formations interchangeability.

**Property 1** (Strict Second-Order Additivity). *The expected model distortion $\Gamma(\boldsymbol{O}, \widehat{\boldsymbol{O}})$ between the output of a pretrained model $\boldsymbol{O}$ and a quantized one $\widehat{\boldsymbol{O}}$ on dataset $X$ can be approximated into the sum of model distortion due to the quantization of individual layers:*

$$\Gamma(\boldsymbol{O}, \widehat{\boldsymbol{O}}) = \mathbb{E}_X(\|\boldsymbol{O} - \widehat{\boldsymbol{O}}\|_F^2) \approx \sum_l \Gamma_{\widehat{\boldsymbol{W}}_l}(\boldsymbol{O}, \widehat{\boldsymbol{O}}) + \sum_l \Gamma_{\widehat{\boldsymbol{A}}_l}(\boldsymbol{O}, \widehat{\boldsymbol{O}}), \tag{12}$$

Recall that $\Delta \boldsymbol{W}_l$ and $\Delta \boldsymbol{A}_l$ are the perturbation on layer weights and activations respectively by quantization, *i.e.*, $\Delta \boldsymbol{W}_l = \boldsymbol{W}_l - \widehat{\boldsymbol{W}}_l, \Delta \boldsymbol{A}_l = \boldsymbol{A}_l - \widehat{\boldsymbol{A}}_l$. The derivation of Property 1 utilizes a basic assumption as follows:

**Assumption 1.** *Quantization perturbation Layer-independency Zhou et al. (2018): The perturbation on weight/activation by individual quantizing on each layer is (1) zero-meaned:* $\forall 1 \leqslant i \leqslant L, \mathbb{E}(\Delta W_i) = \mathbb{E}(\Delta A_i) = 0$, *and (2) independent across layer:* $\forall 1 \leqslant i \neq j \leqslant L, \mathbb{E}(\Delta W_i)\mathbb{E}(\Delta W_j) = \mathbb{E}(\Delta A_i)\mathbb{E}(\Delta A_j) = \mathbb{E}(\Delta W_i)\mathbb{E}(\Delta A_j) = 0.$

We provide more discussion and evidences in Appendix A.1.

*Proof.* According to second-order taylor series expansion when under perturbations on $l$-th layer's weight (first order term can be neglected on pretrained models):

$$\hat{O} \approx O + \frac{1}{2}\Delta W_l^\top H_l^w \Delta W_l, \tag{13}$$

The L2 distortion when all the layer weights are perturbated can be written as

$$\Gamma_{\widehat{W}}(O, \hat{O}) = \mathbb{E}_X\left[\|O - \hat{O}\|_F^2\right] \approx \mathbb{E}_X\left[\left(\sum_{i=1}^l \frac{1}{2}\Delta W_i^\top H_i^w \Delta W_i\right)^\top \left(\sum_{j=1}^l \frac{1}{2}\Delta W_j^\top H_j^w \Delta W_j\right)\right]$$

$$= \mathbb{E}_X\left[\sum_{i,j=1}^l \left(\frac{1}{2}\Delta W_i^\top H_i^w \Delta W_i\right)^\top \left(\frac{1}{2}\Delta W_j^\top H_j^w \Delta W_j\right)\right]. \tag{14}$$

The summation inside expectation can be further swapped out with the expectation:

$$\Gamma_{\widehat{W}}(O, \hat{O}) \approx \sum_{i,j=1}^l \mathbb{E}_X[\frac{1}{4}\Delta W_i H_i^{w\top} \Delta W_i \Delta W_j^\top H_j^w \Delta W_j]. \tag{15}$$

Further utilize independency assumption in Assumption 1 on the expectation on matrices: (replace $\mathbb{E}_X$ with $\mathbf{E}$ for visual simplicity)

$$\Gamma_{\widehat{W}}(O, \hat{O}) \approx \frac{1}{4}\mathbf{E}[\Delta W_i]H_i^{w\top}\mathbf{E}[\Delta W_i \Delta W_j^\top]H_j^w\mathbf{E}[\Delta W_j]. \tag{16}$$

Notice that (1) $\mathbf{E}[\Delta W_i]$ and $\mathbf{E}[\Delta W_j]$ in the above complies with the zero-mean assumption when $i \neq j$ in Assumption 1, (2) $\mathbf{E}[\Delta W_i \Delta W_j^\top] = 0$ since assumption 1 indicates $\Delta W_i$ and $\Delta W_j$ are independent, and (3) the fact that all-zero matrix multiplied with any matrix results in zero, we can eliminate such cross terms $i \neq j$ in the summands, resulting in:

$$\Gamma_{\widehat{W}}(O, \hat{O}) \approx \sum_{i=1}^l E\left(\left\|\frac{1}{2}\Delta W_i^\top H_i^w \Delta W_i\right\|^2\right). \tag{17}$$

The above derivation also applies to the case of activation quantization, resulting:

$$\Gamma_{\widehat{A}}(O, \hat{O}) \approx \sum_{i=1}^l E\left(\left\|\frac{1}{2}\Delta A_i^\top H_i^a \Delta A_i\right\|^2\right). \tag{18}$$

Therefore, the original expected L2 distortion under weights and activation perturbation is:

$$\Gamma(O, \hat{O}) \approx \sum_{i=1}^l \Gamma_{\widehat{W}_l}(O, \hat{O}) + \sum_{i=1}^l \Gamma_{\widehat{A}_l}(O, \hat{O})$$

$$= \sum_{i=1}^l E\left(\left\|\frac{1}{2}\Delta W_i^\top H_i^w \Delta W_i\right\|^2\right) + \sum_{i=1}^l E\left(\left\|\frac{1}{2}\Delta A_i^\top H_i^a \Delta A_i\right\|^2\right). \tag{19}$$

$\square$

**Property 1.1** (Upper-bound Second-Order Additivity). *The expected model distortion $\Gamma(O, \hat{O})$ between the output of a pretrained model $O$ and a quantized one $\hat{O}$ on dataset $X$ can be upper bounded by the sum of model distortion due to the quantization of individual layers:*

$$\Gamma(O, \hat{O}) = \mathbb{E}_X(\|O - \hat{O}\|_F) \leqslant \sum_l \Gamma_{\widehat{W}_l}(O, \hat{O}) + \sum_l \Gamma_{\widehat{A}_l}(O, \hat{O}), \tag{20}$$

*Proof.* According to second-order taylor series expansion when under perturbations on weights and activations (first order term can be neglected on pretrained models):

$$\hat{O} \approx O + \sum_{l=1}^{L} \frac{1}{2} \Delta W_l^\top H_l^w \Delta W_l + \sum_{l=1}^{L} \frac{1}{2} \Delta A_l^\top H_l^a \Delta A_l, \tag{21}$$

The L2 distortion when all the layer weights are perturbated can be written as

$$\Gamma_{\widehat{W}}(O, \hat{O}) = \mathbb{E}_X \left[ \|O - \hat{O}\|_F \right] \approx \mathbb{E} \left[ \left\| \sum_{l=1}^{L} \frac{1}{2} \Delta W_l^\top H_l^w \Delta W_l + \sum_{l=1}^{L} \frac{1}{2} \Delta A_l^\top H_l^a \Delta A_l \right\|_F \right] \tag{22}$$

Then we can apply the triangular inequality property of the L1-norms of two variables $\sum_l \frac{1}{2} \Delta W_l^\top H_l^w \Delta W_l$ and $\sum_l \frac{1}{2} \Delta A_l^\top H_l^a \Delta A_l$, giving:

$$\| \sum_{l=1}^{L} \frac{1}{2} \Delta W_l^\top H_l^w \Delta W_l + \sum_{l=1}^{L} \frac{1}{2} \Delta A_l^\top H_l^a \Delta A_l \| \leqslant \sum_{l=1}^{L} \| \frac{1}{2} \Delta W_l^\top H_l^w \Delta W_l \| + \sum_{l=1}^{L} \| \frac{1}{2} \Delta A_l^\top H_l^a \Delta A_l \|. \tag{23}$$

Therefore we have $\Gamma(O, \hat{O}) \leqslant \mathbb{E}(\sum_l \| \frac{1}{2} \Delta W_l^\top H_l^w \Delta W_l \|) + \mathbb{E}(\sum_l \| \frac{1}{2} \Delta A_l^\top H_l^a \Delta A_l \|)$. Since summations inside expectation can be swapped out of expectation, we have

$$\Gamma(O, \hat{O}) \leqslant \sum_l \mathbb{E}(\|\Delta W_l^\top H_l^w \Delta W_l\|_F) + \sum_l \mathbb{E}(\|\Delta A_l^\top H_l^a \Delta A_l\|_F),$$

that is

$$\Gamma(O, \hat{O}) \leqslant \sum_l \Gamma_{\hat{W}_l}(O, \hat{O}) + \sum_l \Gamma_{\hat{A}_l}(O, \hat{O}). \tag{24}$$

$\square$

## A.1 Evidences of Assumption Used

We now show the empirical observations on real ViT models to further support the assumptions used in the above theoretical proof.

**Assumption 1 (1): Zero-mean:** Fig. 2 illustrates the distributions of weight perturbation $\Delta W$ in different layers of different models with different bitwidths. Under majority circumstances, the distributions of the element-wise perturbations display a symmetric pattern around zero, supporting that zero-mean assumption holds universally for ViTs.

**Assumption 1 (2): Interlayer independency:** Interlayer independency can be evaluated by observing the covariance between $\Delta W_i$ and $\Delta W_j$. Fig. 4 illustrates the the full pair-wise combination of all quantizable layers in the model, where each entry is the layerwise expectation of product of weight perturbation of two layers averaged on a random batch of images from ImageNet. It displays a clear diagonal pattern that the covariances vanish on off-diagonal locations where $i \neq j$, which is observed on all 4 ViTs. This supports the interlayer-independence assumption.

**Independency between same layer weight and activation.** We also provide empirical studies showing the correlation coefficients and the covariance behaviors for the quantization errors between the same layer weight and activation. As shown in Fig. 5, we observe that the covariance between quantization error of weight and activation has significantly smaller magnitude compared to self-variance of the weight and activation, by at least $10^2$ for all the layers. This shows the weights and activations from same layer satisfies the independence assumption on the quantization error.

## A.2 Evidences of Additivity

To further support the theoretically discovered Property 1 (Section A), we further provide direct empirical evidences on ViT models as in Fig. 6 and Fig. 7 at low bit width weight/activation quantization from 5 bits down to 2 bits. As one can see, at 4 to 5 bits, most of points lie in the diagonal meaning that the additivity property holds well. However, when the bit width is less than 4 bits, outlier points start to increase. In practice, our approach can quantize the model into 6 bits to 8 bits without hurting accuracy. There is a noticeable accuracy drop when the model is quantized to lower bit width (e.g, 4 bits or less).

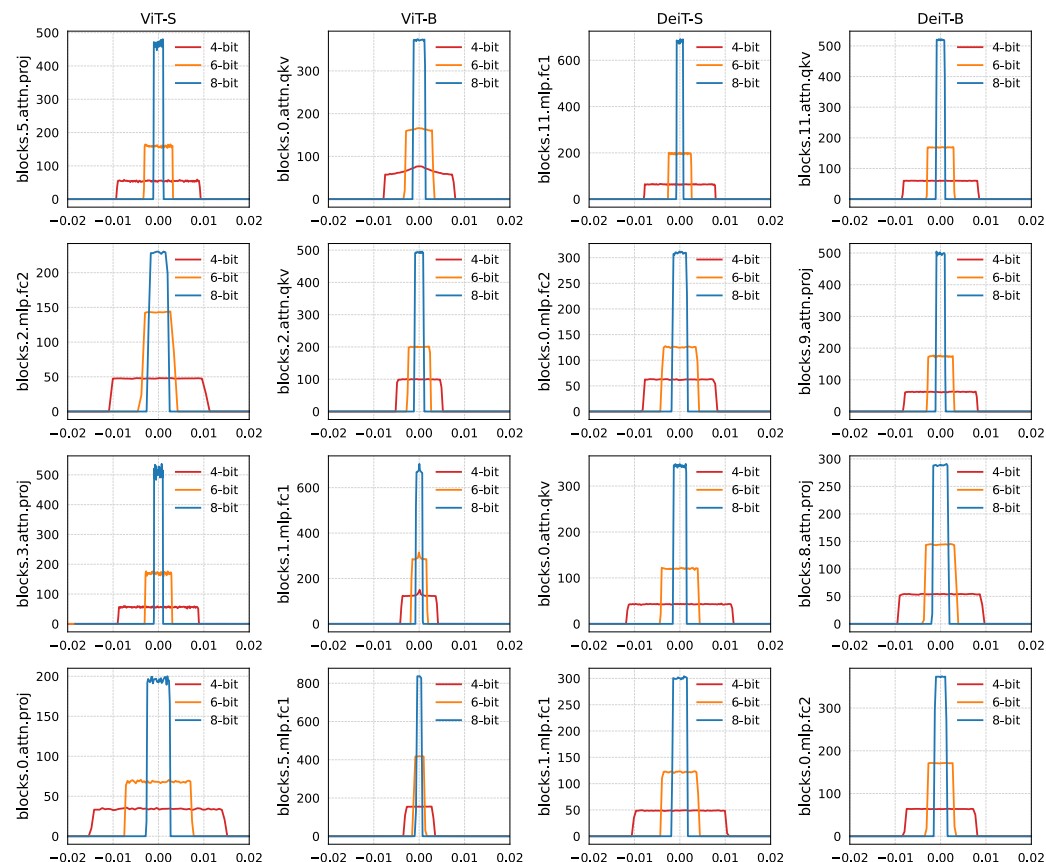

Figure 2: Empirical statistics of histogram of weight perturbation $\Delta \boldsymbol{W}_i$ when quantizing individual layers. We randomly select 4 layers from each models to display the distribution quantization error matrices under various bit-widths. We use the same symmetric uniform quantizer with least MSE scaling factors.

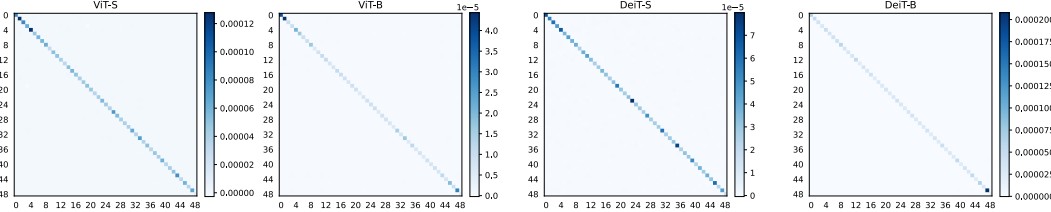

Figure 3: Empirical statistics of $E(\|\Delta \boldsymbol{W}_i \Delta \boldsymbol{W}_j\|^2)$ when quantizing individual layers in pretrained model. Deeper blue color indicates larger value, vice versa.

## B  RELATION BETWEEN MODEL DISTORTION AND MODEL ACCURACY

Our BLOB-Q method optimizes the mixed-precision bit-allocation directly on model distortion, i.e. the L2 distortion on model output between FP and quantized model. This is based on the hypothesis that model distortion directly reflects the task performance. To evaluate this hypothesis, we evaluate this correlation on empirical models in Fig. 8. As it shows, the model distortion is closely correlated to accuracy under various model bit-rates under 3 models, where the large distortion is associated with poor model accuracy. This strongly supports the hypothesis that is also one of the motivations of BLOB-Q.

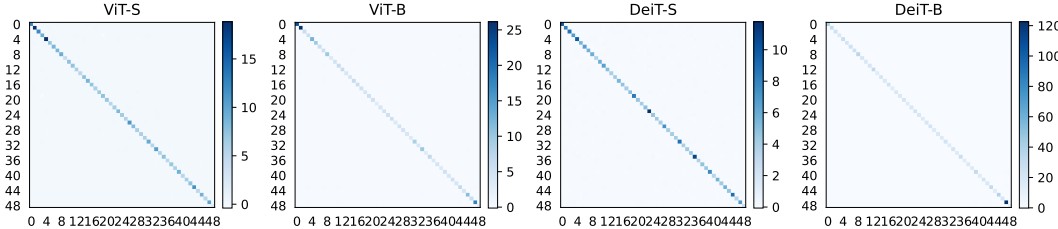

Figure 4: Empirical statistics of covariance matrix $E(\Delta \boldsymbol{W}_i \Delta \boldsymbol{W}_j)$ when quantizing individual layers in pretrained model. Deeper blue color indicates larger value, vice versa.

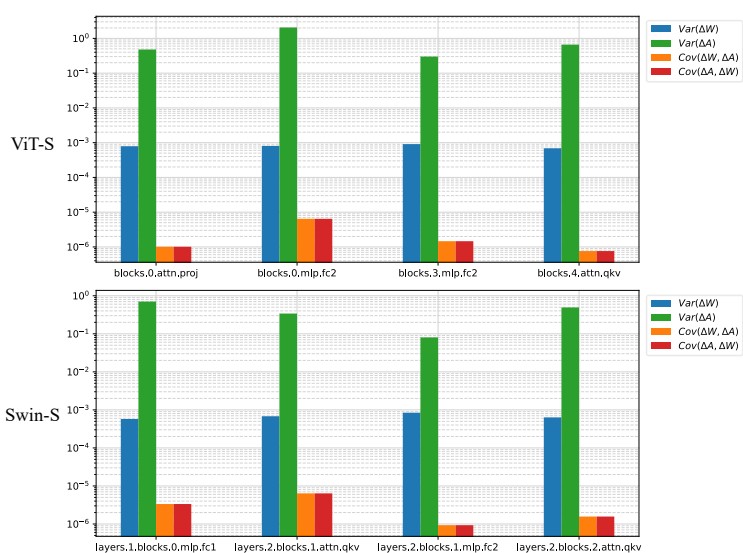

Figure 5: Covariance statistics of quantization error on same layer weight and activation. We randomly sample four layers from each models and quantize its weight/activation to 2-bit. Y-axis are plotted in log-scale.

## C  PARALLEL DP-SOLVER DESIGN

A naive DP-solver can be implemented based on Section 4 which by default performs sequentially on CPU, but we can harness the parallel-computing power of GPU to further accelerate the theoretical time cost to enable DP-Solver for larger models at scale. DP-solver to solve the integer problem which can be regarded as finding the minimally achieveable bitwidth ("weight") for each group of to-be-quantized layer weight/activation ("item"). In the first step, since from Eq. 10 we can see that the states S[l,:] only depends on previous layer S[l-1,:], and the fact that the final solutions are only related to G, thus we can reduce the memory cost of the outer-most loop (Algorithm 1 L13) by allocate a smaller 2-by-$R$ matrix for S to reduce the memory allocation by $L/2$ times. This avoids OOM issue on large models like DeiT-B/384 with higher target rates *e.g.* 8-bit. Secondly, in the states updating in DP-solver (Algorithm 1 L13-17), the number of states needed to iterate the inner-loop can be actually reduced to $\sum_i^l d_i B \leqslant R$ as this is the largest reachable accumulated sizes when all visited layers (from 1-st to $l$-th) quantized to the maximum bitwidth $B$. This can reduces the complexity for the nested for-loop by virtually half. Also, to avoid OOM for large layer sizes, we perform parallel state updating piecewisely for the inner-loop, allowing us to adjust the piece length to fully utilize VRAM. Finally, for the matrix G, we can offload it to storage layerwisely for G[l,:] since as Eq. 11 indicates it does not depend on G of other layers, and we can load back to perform Algorithm 1 L19-23. However, for large models *e.g.* DeiT-B with target bit-rate of 8-bit, the storage needed for offloaded G can be as large as a long matrix of $50 \times 8.6 \times 10^7 \times 8 \times 8 \text{bytes} \approx 256\text{GB}$, which is unacceptable. We notice that due to the discrete behavior of the state transfer, at each layer, a lot of entries in G repeat themselves for as long as $d_l$. Therefore we perform run-length-coding to encode G before offloading to storage and efficiently decode it back for the final steps of the algorithm.

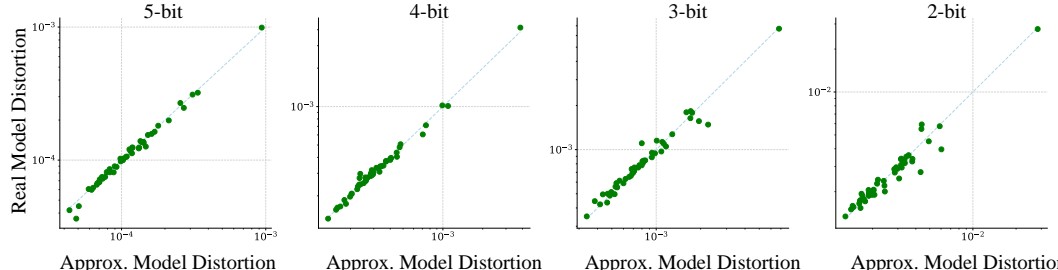

Figure 6: Empirical Evidences of Second-order additivity property on ViT-S. The X-axis represents the model distortion when approximating via second-order additivity property (Property 1) when quantizing all layer **weights**. The Y-axis represents the real model distortion by directly measuring the L2 distortion on model output. Data points inclines to the diagonal lines very well, showing that the additivity property holds.

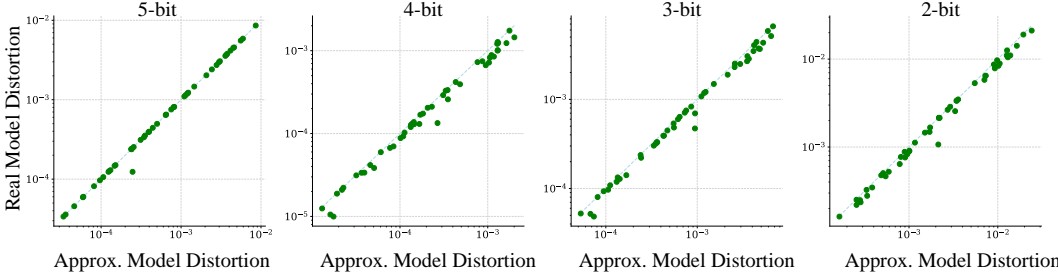

Figure 7: Empirical Evidences of Second-order additivity property on DeiT-B. The X-axis represents the model distortion when approximating via second-order additivity property (Property 1) when quantizing all layer **activations**. The Y-axis represents the real model distortion by directly measuring the L2 distortion on model output. Data points inclines to the diagonal lines very well, showing that the additivity property holds.

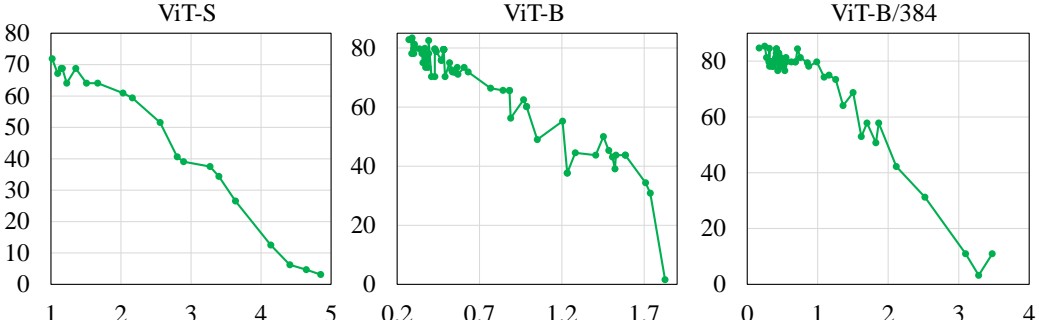

Figure 8: Model Distortion (X-axis) vs. Model Accuracy (Y-axis). We quantize each model to decreasing target mixed-precision bit-widths using BLOB-Q. We evaluate the average L2 distortion on 128 calibration images from ImageNet training set.

Empirically this significantly reduces the encoded size of G to only around 20MiB for DeiT-B model. The attached source code contains the complete details of the proof-of-concept implementation of the above described parallel DP-solver.

# D  ADDITIONAL EXPERIMENT RESULTS

## D.1  RESULTS WITH FINETUNING

Although BLOB-Q does not require finetuning to compete with plenty of SOTA methods as in Tab. 1 in main text, for completeness we investigate the effect of finetuning on top of raw BLOB-Q results.

Table 6: Additional Finetuning results after Post-training Quantization (marked as "**+ FT**"). Top-1 classification accuracy is evaluated. MP denotes quantization with Mixed-Precision. For a fair comparison, we report the results with the same average bit widths. For example, 4 MP means that we quantize the ViT Models into 4 bits on average across the layers. The best result of each group is highlighted with bold.

| Method | Size (Bit) | ViT-S | ViT-B | ViT-B/384 | DeiT-S | DeiT-B | DeiT-B/384 |
|---|---|---|---|---|---|---|---|
| Full Precision | 32 | 81.39% | 84.54% | 86.05% | 79.87% | 81.85% | 83.12% |
| PTQ4ViT Yuan et al. (2022) | 4 | 42.57% | 30.69% | - | 34.08% | 64.39% | - |
| APQ-ViT Ding et al. (2022) | 4 | 47.95% | 41.41% | - | 43.55% | 67.48% | - |
| RepQ-ViT Li et al. (2023) | 4 | 65.60% | 68.48% | - | 69.03% | 75.61% | - |
| BLOB-Q | 4 MP | 67.32% | 79.97% | 71.20% | 75.18% | 79.48% | 80.17% |
| BLOB-Q + FT | 4 MP | **68.53%** | **80.66%** | **72.39%** | **77.42%** | **80.15%** | **81.0%** |
| PTQ Liu et al. (2021b) | 6 MP | 70.24% | 75.26% | 46.88% | 75.10% | 77.47% | 68.44% |
| PTQ4ViT Yuan et al. (2022) | 6 | 78.63% | 81.65% | 83.34% | 76.28% | 80.25% | 81.55% |
| Percentile Li et al. (2019) | 6 MP | 67.74% | 77.63% | 77.60% | 70.49% | 73.99% | 78.24% |
| EasyQuant Wu et al. (2020) | 6 | 75.13% | 81.42% | 82.02% | 73.26% | 75.86% | 81.26% |
| APQ-ViT Ding et al. (2022) | 6 | 79.10% | 82.21% | - | 77.76% | 80.42% | - |
| RepQ-ViT Li et al. (2023) | 6 | 80.43% | 83.62% | - | 77.76% | 80.42% | - |
| NoisyQuant Liu et al. (2023b) | 6 | 78.65% | 82.32% | 83.22% | 77.43% | 80.70% | 81.65% |
| BLOB-Q | 6 MP | **80.53%** | 83.66% | **85.20%** | 79.22% | 81.58% | 82.72% |
| BLOB-Q + FT | 6 MP | 80.31% | **84.23%** | 85.15% | **79.53%** | **81.85%** | **82.98%** |
| PTQ Liu et al. (2021b) | 8 MP | 80.46% | 76.98% | 85.35% | - | 75.94% | - |
| PTQ4ViT Yuan et al. (2022) | 8 | 81.00% | 84.25% | 85.82% | 79.47% | 81.48% | 82.97% |
| Percentile Li et al. (2019) | 8 MP | 78.77% | 80.12% | 82.53% | 73.98% | 75.21% | 80.02% |
| EasyQuant Wu et al. (2020) | 8 | 80.75% | 83.89% | 85.53% | 76.59% | 79.36% | 82.10% |
| APQ-ViT Ding et al. (2022) | 8 | 81.25% | 84.26% | - | 79.78% | 81.72% | - |
| NoisyQuant Liu et al. (2023b) | 8 | 81.15% | 84.22% | 85.86% | 79.51% | 81.45% | 82.49% |
| I-ViT Li & Gu (2023) | 8 | 81.27% | 84.76% | - | **80.12%** | 81.74% | - |
| FQ-ViT Lin et al. (2021) | 8 | - | 83.31% | - | 79.17% | 81.20% | - |
| BLOB-Q | 8 MP | **81.54%** | 84.84% | 85.93% | 79.59% | 81.75% | 82.85% |
| BLOB-Q + FT | 8 MP | 81.47% | **84.89%** | **86.01%** | 79.79% | **81.89%** | **83.04%** |

As illustrated in Tab. 6, we can see that finetuning burings noticeable gain from the retraining-free results on all 6 models in most cases, especially the case under low bit rate like 4-bit MP. On higher bit rates (6/8-bit), given the already small performance gaps from dense model, finetuning brings small improvements for most cases, and maintains similar performances of retraining-free results for few others. We currently finetune for only 100 iterations on ImageNet training set. With dedicated hyperparameter tuning, e.g. with larger epochs and other optimizer choices, we may see even more performance gain.

## D.2 SCALABILITY OF LOWER QUANTIZATION BIT-WIDTH

Beyond the 4-bit quantization that most existing methods, we further evaluated the quantization results on even lower bit-width (3-bit), and we found more significant improvement on 3-bit compared to others, up to 41.15% on ViT-L as shown in Tab. 7. As claimed in our paper, one of our main contributions of this paper is to boost the performance of ViT quantization on low bit-width. This is useful in practice when the model is required to be quantized to very small sizes for deployment.

Table 7: Extended experiments on under 3-bit. Both weights and activations are quantized.

| Method | Bit-width | ViT-B | ViT-L |
|---|---|---|---|
| NoisyQuant Liu et al. (2023b) | 3 | 0.138 | 1.72 |
| RepQ-ViT Li et al. (2023) | 3 | 0.27 | 0.85 |
| Ours | 3 MP | **36.36** | **42.87** |

## D.3 BIT ALLOCATION STATISTICS

We some example visualizations of the bit allocations resulted from BLOB-Q on ViTs. Fig. 9 and Fig. 10 show the distribution of the avagerged bits allocated blockwisely inside each layer under two

ViT models, ViT-S and ViT-B, and quantized to 4 and 6-bits respectively. We notice some patterns for the allocation results. Firstly, aligning with previous observations in many compression works that first and last layers are usually sensitive to quantization thus requires more bits, we also observe similar phenomenon on ViTs especially for activation quantization. Beside that, we also found an interesting pattern that the `fc1` layers in latter transformer blocks requires slightly more bits to quantize their activations, which is constantly observed under multiple settings.

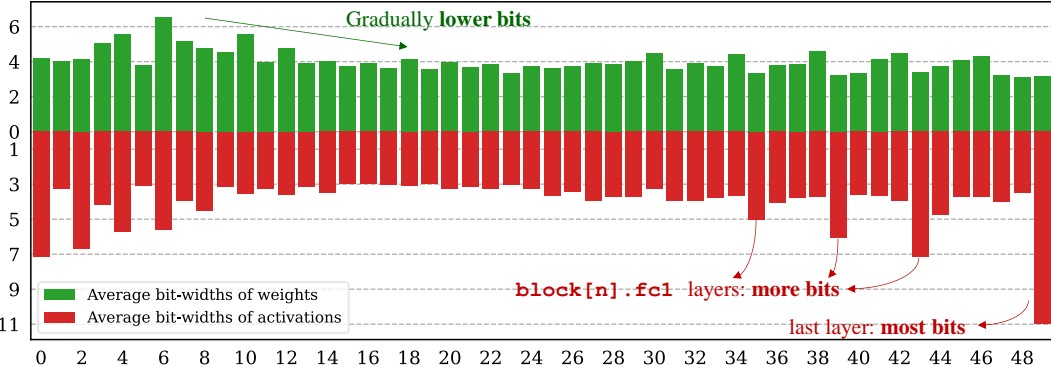

Figure 9: Bit allocation visualization of BLOB-Q on 4-bit ViT-B. We show the averages of bit rates allocated to each block of weight/activation tensor within each layer.

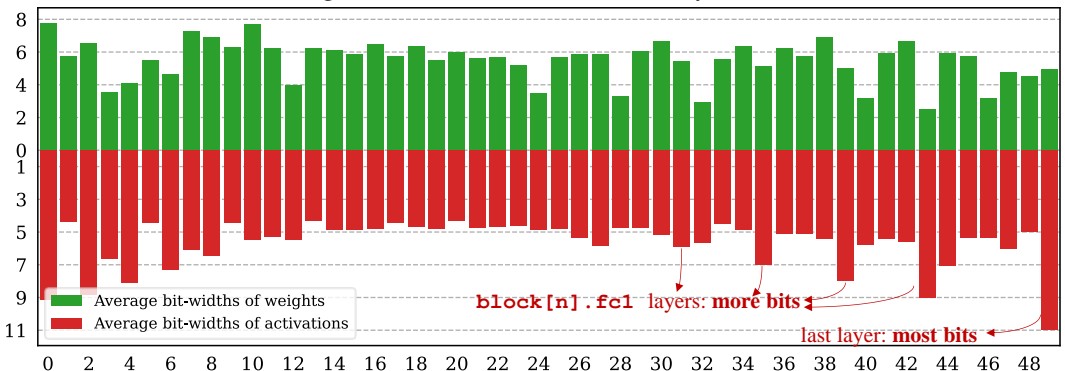

Figure 10: Bit allocation visualization of BLOB-Q on 6-bit ViT-B. We show the averages of bit rates allocated to each block of weight/activation tensor within each layer.

## D.4 EMPIRICAL TIME AND MEMORY COSTS

Fig. 11 shows the time costs of our proposed BLOB-Q compared to the most competitive methods to us. BLOB-Q consists of (1) **BLOB-Q-Calib** a one-time calibration on weight/activation quantization which can be used to solve MPQ with arbitrary target rates, and (2) **BLOB-Q-DP** the DP-solver to get final solution for a specific bit-width target (*e.g.* 4/6/8-bit). As the figure shows, our method displays minimal optimization times (BLOB-Q-DP) and significantly faster than NoisyQuant Liu et al. (2023b). Moreover, the total times (BLOB-Q-Total) combining two steps together still remain competitive within manageable range. Note that in practice, the calibration is only required to perform once and versatile to arbitrary target bit rates, therefore the calibration time (BLOB-Q-Calib) does not significantly contribute to overall time consumption for *e.g.* multi-target scenario.

Fig. 12 shows the memory costs of calibrating ViT models on L40 machine. As it shows, the VRAM memory required to perform PTQ ranges reasonably between around 2GB and 6GB for the evaluated models, making our BLOB-Q accessible to more use cases.

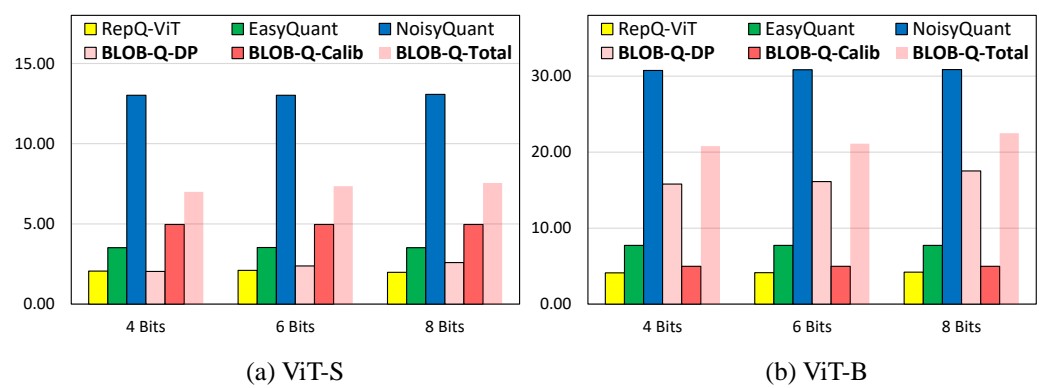

(a) ViT-S                                        (b) ViT-B

Figure 11: Time costs (minutes) of different PTQ methods.

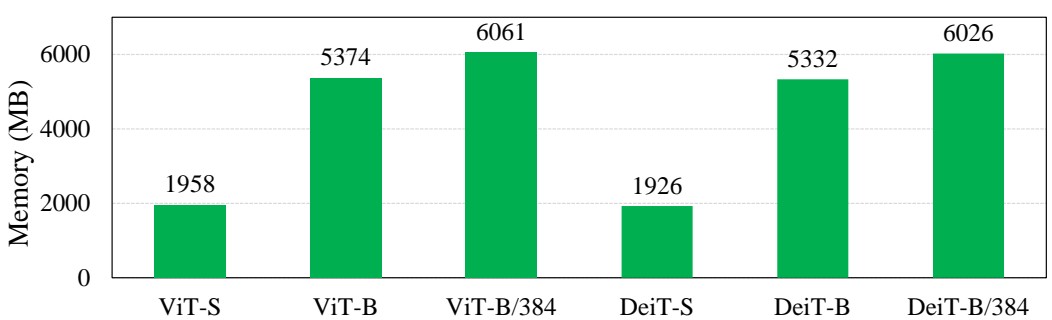

Figure 12: VRAM Memory consumptions (MB) of BLOB-Q calibration.

### D.5  MORE VIT VARIANTS AND NON-TRANSFORMER MODELS

We further evaluate the scalability of our method to other ViT variants, including ViT-L and Swin variants like Swin-T and Swin-S, as well as pre-ViT architectures like ResNet-18. As shown in Tab. 8, we observe consistent advantage compared to baseline method on various bit-widths, especically for lower bit-widths where we improve from baseline method by up to 11.24% under 4-bit. This shows that the effectiveness of the BLOB-Q framework is not constrained to ViT architecture.

Table 8: More ViT variants and CNNs results on ImageNet-1k .

| Method | W-bit | A-bit | ViT-L | Swin-T | Swin-S | ResNet-18 |
|---|---|---|---|---|---|---|
| Full Precision | 32 | 32 | 85.59 | 81.2 | 83.23 | 71.61 |
| BLOB-Q | 8 MP | 8 MP | 85.82 | 81.12 | 83.11 | 71.74 |
| BLOB-Q | 6 MP | 6 MP | 85.77 | 80.91 | 82.82 | 70.23 |
| BLOB-Q | 4 MP | 4 MP | 77.09 | 76.41 | 80.53 | 63.02 |

### D.6  COMPARISON WITH UNIFORM BASELINES UNDER MIXED-PRECISION SETTING

To faciliate a fairer comparison with uniform-precision baselines, we integrated a SOTA mixed-precision method into various uniform-precision baselines to enable a fairer comparison. Specifically, we choose a latest mixed-precision method $P^2$-ViT to perform integration. We conduct the integration for as many as uniform-precision PTQ baselines with published code available, i.e., RepQ-ViT, EasyQuant, NoisyQuant and PTQ4ViT. We integrate each of their quantizer into $P^2$-ViT evolutionary bit searching process to generate corresponding mixed-precision results. We show the full comparison under mixed-precision setting in Tab. 9. Firstly, we compare the mixed-precision setting and uniform-precision setting for 4 uniform-precision methods mentioned above. We notice that the mixed-

precision results of these uniform-precision methods still underperform our method. Under 4-bit, our method outperforms RepQ-ViT, EasyQuant, NoisyQuant and PTQ4ViT by at least 2.67%, 8.65%, 7.44% and 18.01% respectively. Under 8-bit and 6-bit where bit-widths are high enough, we also notice similar improvement. This verifies that the superior results of our method indeed come from the proposed mixed-precision approach rather than adopting mixed-precision scheme itself.

Table 9: Comparisons under mixed-precision settings, Method + $P^2$-ViT represent the allocating mixed-precisioned bit-widths with $P^2$-ViT bit searching on top of each respective method.

| Method | Bit | ViT-S | ViT-B | DeiT-S | DeiT-B |
|---|---|---|---|---|---|
| Full Precision | 32 | 81.39 | 84.54 | 79.87 | 81.85 |
| RepQ-ViT | 4 | 64.43 | 66.27 | 67.67 | 75.80 |
| RepQ-ViT + $P^2$-ViT | 4 MP | 64.65 | 67.08 | 68.15 | 79.18 |
| EasyQuant | 4 | 34.94 | 71.03 | 28.15 | 68.85 |
| EasyQuant + $P^2$-ViT | 4 MP | 37.21 | 71.32 | 30.08 | 69.13 |
| NoisyQuant | 4 | 50.36 | 69.38 | 38.46 | 72.84 |
| NoisyQuant + $P^2$-ViT | 4 MP | 51.35 | 69.73 | 38.63 | 72.86 |
| PTQ4ViT | 4 | 40.73 | 55.33 | 27.01 | 61.42 |
| PTQ4ViT + $P^2$-ViT | 4 MP | 40.81 | 60.73 | 29.64 | 62.29 |
| **Ours MP** | **4 MP** | **67.32** | **79.97** | **77.42** | **80.30** |
| RepQ-ViT | 6 | 80.14 | 83.39 | 78.69 | 81.16 |
| RepQ-ViT + $P^2$-ViT | 6 MP | 80.15 | 83.41 | 78.81 | 81.23 |
| EasyQuant | 6 | 77.27 | 81.54 | 65.08 | 79.34 |
| EasyQuant + $P^2$-ViT | 6 MP | 77.38 | 82.85 | 67.93 | 79.55 |
| NoisyQuant | 6 | 77.72 | 82.10 | 65.12 | 79.67 |
| NoisyQuant + $P^2$-ViT | 6 MP | 77.75 | 82.48 | 67.59 | 79.74 |
| PTQ4ViT | 6 | 78.02 | 82.90 | 76.29 | 80.19 |
| PTQ4ViT + $P^2$-ViT | 6 MP | 78.53 | 83.09 | 76.37 | 80.29 |
| **Ours MP** | **6 MP** | **80.73** | **84.23** | **79.53** | **81.85** |
| RepQ-ViT | 8 | 81.23 | 84.43 | 78.69 | 81.16 |
| RepQ-ViT + $P^2$-ViT | 8 MP | 81.23 | 84.49 | 79.69 | 81.74 |
| EasyQuant | 8 | 80.87 | 84.77 | 78.79 | 81.34 |
| EasyQuant + $P^2$-ViT | 8 MP | 80.96 | 84.80 | 78.87 | 81.37 |
| NoisyQuant | 8 | 80.92 | 84.76 | 78.89 | 81.32 |
| NoisyQuant + $P^2$-ViT | 8 MP | 81.02 | 84.84 | 79.05 | 81.40 |
| PTQ4ViT | 8 | 80.90 | 84.63 | 79.43 | 81.48 |
| PTQ4ViT + $P^2$-ViT | 8 MP | 80.93 | 84.70 | 79.53 | 81.54 |
| **Ours MP** | **8 MP** | **81.54** | **84.89** | **81.25** | **81.75** |

## D.7 INTEGRATING WITH OTHER MIXED-PRECISION METHODS

To evaluate the performance of the bit-allocation strategies in a controlled setting, we evaluated the performance when integrating the bit-width allocations of 2 recent SOTA mixed-precision methods respectively into our approach, including EMQ Dong et al. (2023) and OMPQ Ma et al. (2023), and then evaluated their results. Specifically, we replace our original bit-allocation with theirs once at a time, while keeping other parts of our approach the same (e.g. quantizer choices). As shown in Tab. 10, we compare the performances of our method as well as when we combine the other two bit allocation strategies. "EMQ/OMPQ + Our quantizers" denotes the results when we replace the bit-allocation strategies of EMQ/OMPQ respectively with our bit allocation strategy; "Our bit allocation + Our quantizers" denotes our method. We observe that our method outperforms both "EMQ + Our quantizers" and "OMPQ + Our quantizers". Under 4-bit, we observe significant accuracy discrepancies by at least 10.29% and 13.19% from EMQ and OMPQ respectively. Under 6/8-bit, we observe similar accuracy discrepancies between the integration results and our original results. This shows that our bit allocation strategy is indeed stronger than existing bit-allocation strategies.

Table 10: Integration results with other mixed-precision methods with our quantizers.

| Method | Bit | ViT-S | ViT-B | DeiT-S | DeiT-B |
|---|---|---|---|---|---|
| EMQ + Our quantizers | 4 MP | 57.03 | 70.28 | 44.72 | 69.71 |
| OMPQ + Our quantizers | 4 MP | 40.04 | 66.13 | 41.17 | 67.11 |
| Our bit allocation + Our quantizers | 4 MP | **67.32** | **79.97** | **77.42** | **80.30** |
| EMQ + Our quantizers | 6 MP | 79.30 | 82.78 | 73.47 | 78.37 |
| OMPQ + Our quantizers | 6 MP | 78.24 | 82.37 | 73.23 | 77.91 |
| Our bit allocation + Our quantizers | 6 MP | **80.73** | **84.23** | **79.53** | **81.85** |
| EMQ + Our quantizers | 8 MP | 81.36 | 84.86 | 76.08 | 80.19 |
| OMPQ + Our quantizers | 8 MP | 81.13 | 84.74 | 75.96 | 80.08 |
| Our bit allocation + Our quantizers | 8 MP | **81.54** | **84.89** | **81.25** | **81.75** |

## D.8 COMPARSION OF DYNAMIC PROGRAMMING AND PARETO METHOD

We have conducted additional studies to verify the superiority of the dynamic programming. Specifically, we replace the dynamic programming algorithm with Pareto or linear programming, and then compare the results. We implement the Lagrangian algorithm to linearly find the pareto frontier given a Lagrangian multiplier, which determines the slope of the linear approacher. As the Tab. 11 shows, we compare the performance of Pareto method to our dynamic programming-based method under 4/6/8-bits. We observe that the proposed dynamic programming solution consistently outperforms linear Pareto method by up to 62.5%, 11.78% and 4.54% for 4/6/8-bit respectively. This evaluates that the proposed dynamic programming solver indeed finds better solution than local-optimal algorithms.

Table 11: Dynamic Programming v.s. Pareto Method.

| Model | Bit | Pareto Method | Dynamic Programming (Ours) |
|---|---|---|---|
| ViT-S (81.39%) | 4 | 5.07 | **67.32** |
| | 6 | 68.75 | **80.53** |
| | 8 | 79.68 | **81.54** |
| ViT-B (84.54%) | 4 | 72.58 | **79.97** |
| | 6 | 82.81 | **83.66** |
| | 8 | 80.3 | **84.84** |
| ViT-B/384 (86.05%) | 4 | 57.81 | **71.2** |
| | 6 | 80.14 | **85.2** |
| | 8 | 85.3 | **85.93** |
| DeiT-S (79.87%) | 4 | 64.39 | **75.18** |
| | 6 | 78.56 | **79.22** |
| | 8 | 79.73 | **81.25** |
| DeiT-B (81.85%) | 4 | 73.56 | **79.48** |
| | 6 | 79.55 | **81.58** |
| | 8 | 80.21 | **81.75** |
| DeiT-B/384 (83.12%) | 4 | 72.65 | **80.17** |
| | 6 | 81.25 | **82.72** |
| | 8 | 81.71 | **82.85** |

## D.9 SCALABILITY TO LANGUAGE MODELS

Aside from vision tasks, we also explored the performance of BLOB-Q on language models. We first evaluate whether the proposed assumptions and theoretical conclusions are applicable on language

models via similar empirical studies. Then we provide mixed-precision quantization results on those models on various text completion datasets.

**Empirical stuides on assumptions.** We observe similar compliance for the three language models to the proposed zero-mean assumption, inter-layer independency assumption and additivity property, as shown in Fig.13, Fig. 14 and Fig. 15

**Text completion results.** We evaluated on three decoder-only language models, including OPT-350M Zhang et al. (2022), BLOOM-560M AI (2022) and Llama-2-7B Touvron et al. (2023). From the results on text completion task shown in Tab. 12, BLOB-Q also preserve generation quality on wikitext2 dataset with bearable drop in perplexity (PPL). We observe similar trends on the two added dataset as on Wikitext2, where our quantized models only drop by at most 8.87 in perplexity down to 4-bit from full-precision model. This indicates high preservation of language capability for the proposed model. This further evaluates the generalizability of the proposed method on language model. The performance could be further boosted by incorporating more LLM-specific quantizer such as GPTQ [3] quantizer as well as additional finetuning.

Table 12: BLOB-Q on language model OPT-350M results on text completion.

| Model | Method | Bit-width | Wikitext2 PPL ↓ | PTB PPL ↓ | C4 PPL ↓ |
|---|---|---|---|---|---|
| OPT-350M | FP | 32 | 23.5607 | 21.0897 | 21.0934 |
| OPT-350M | BLOB-Q | 8 MP | 25.7557 | 21.0882 | 21.1012 |
| OPT-350M | BLOB-Q | 6 MP | 25.9515 | 21.4077 | 21.3307 |
| OPT-350M | BLOB-Q | 4 MP | 34.6864 | 29.9613 | 27.1915 |
| BLOOM-560M | FP | 32 | 19.4163 | 33.6544 | 25.1162 |
| BLOOM-560M | BLOB-Q | 8 MP | 19.4321 | 33.6591 | 25.126 |
| BLOOM-560M | BLOB-Q | 6 MP | 22.2007 | 40.3169 | 28.2866 |
| Qwen2.5-0.5B | FP | 32 | 11.5153 | 20.4461 | 19.0765 |
| Qwen2.5-0.5B | BLOB-Q | 8 MP | 11.5321 | 20.4937 | 19.1084 |
| Qwen2.5-0.5B | BLOB-Q | 6 MP | 13.5852 | 24.3803 | 22.3268 |

**Text completion case studies.** To facilitate a more intuitive understanding of our performances on the quantized language models, we also provide text completion case studies and show the input prompt and the completed sentences from the quantized models. We list the input prompts and output responses of the quantized models at 8/6/4-bit as well as the full-precision OPT-350M in Fig. 16. We observe that the text completion quality is mostly maintained in quantized model down to 4-bit, in terms of the grammar and semantical coherencies, which is consistent with the perplexity results as shown in the above study in Tab. 12.

### D.10   GENERALIZED RESULTS ON DOWNSTREAM TASK

To evaluate generalizibility of the proposed BLOB-Q beyond standard image classification task, we conduct PTQ experiments on three pretrained Mask-RCNN models with different ViT backbones: Swin-T and Swin-S, obtained by varying training settings such as with multi-scale training data.

We first demonstrate the effectiveness of the proposed BLOB-Q on extremely low bit rates for three ViT-based detection models. We observe a consistent advantage over the uniform quantization method on detection scores across different models and bit rate targets (4 to 6-bit). In Table 13, Table 14, and 15, we achieved near-FP (full-precision) model performance on 6-bit, where three models only drop 0.9–1% on bbox mAP and $< 1\%$ drop on mask mAP from their respective FP models. On 6-bit BLOB-Q induces less detection error compared to the uniform baseline, while on lower bit rates like 5-bit, the performance gap becomes even larger, where we are still able to boost the mAP to the level comparable to FP models. Particularly, we observe a significant boost on extremely low bit such as 4-bit. Compared to the three 4-bit models obtained by uniform quantization where all detection scores are extremely poor, we significantly improve the performances to acceptable ranges using BLOB-Q. Specifically, we improve the bbox mAP score by around $22\times$, from mere 1.4 to 33.0 by employing BLOB-Q. The other two models also show similar improvement.

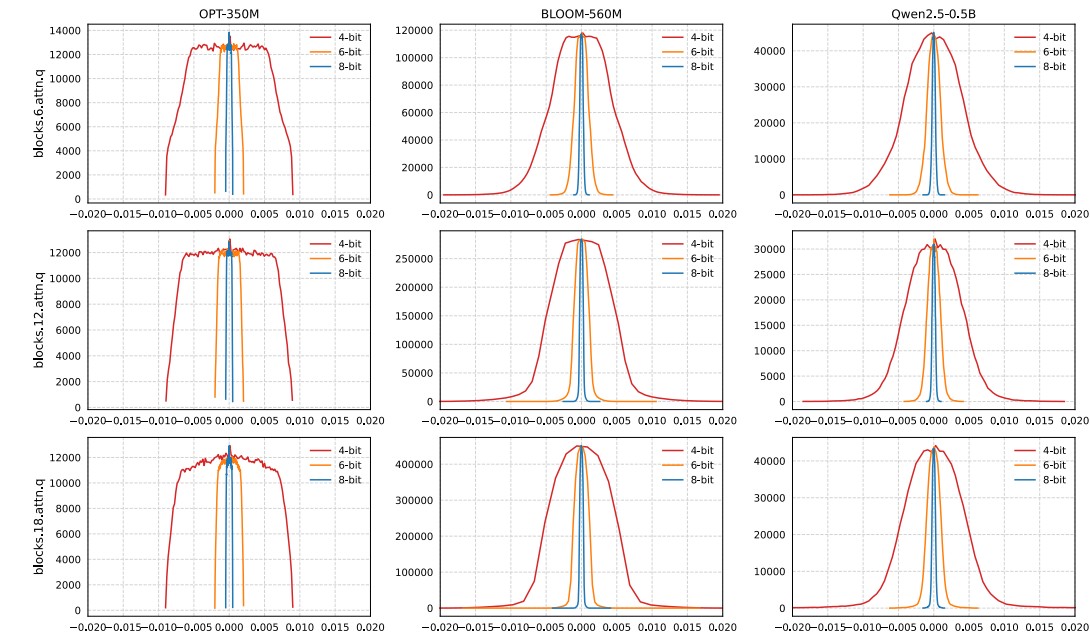

Figure 13: Empirical statistics of histogram of weight perturbation $\Delta \boldsymbol{W}_i$ when quantizing individual layers from three language models. Weights are quantized to 4 bits.

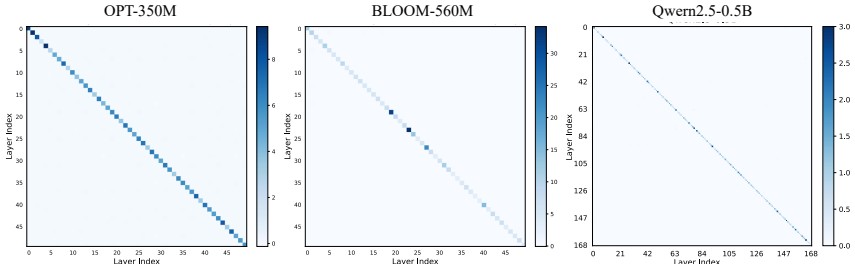

Figure 14: Empirical statistics of $E(\Delta \boldsymbol{W}_i \Delta \boldsymbol{W}_j)$ when quantizing individual layers in three pretrained language models. Deeper blue color indicates larger value, vice versa.

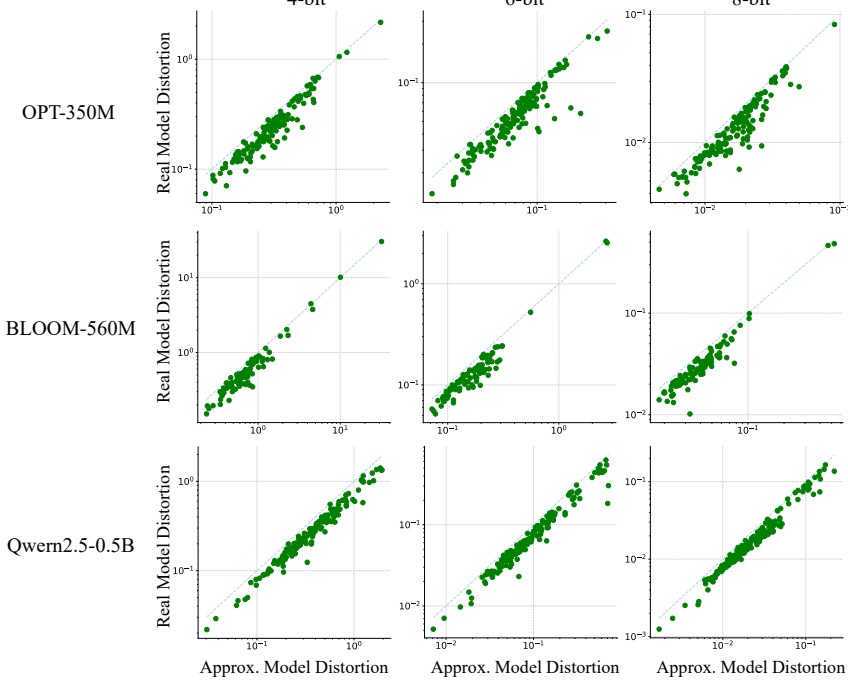

Figure 15: Empirical Evidences of Second-order additivity property on three language models.

| Bit-width | Prompt | Completion |
|---|---|---|
| FP | "Robert Boulter is an English" | "writer and journalist. He is the author of the novel The Last of the Mohicans, which was nominated for the Booker Prize in 2013. He is also the author of the novel The Last of the Mohicans" |
| 8 | "Robert Boulter is an English" | "writer and critic. He is the author of the best-selling novel The Last of the Mohicans, and the forthcoming book The Last of the Mohicans: The Story of the Last of the Mohicans." |
| 6 | "Robert Boulter is an English" | "writer and critic. He is the author of the novel The Last of the Mohicans, which won the National Book Award for Fiction in 2012. He is the author of the novel The Last of the Mohicans" |
| 4 | "Robert Boulter is an English" | "-born, British-based, and Canadian-based, and he is a member of the British Association of Professional Engineers (BAPE).\n\nRobert Boulter is a member of the British Association" |
| FP | "LEWISTON — The Calvary United" | "Methodist Church in Lewiston is hosting a free, family-friendly service on Sunday, Sept. 25, from 10 a.m. to noon.\n\nThe service will be held at the" |
| 8 | "LEWISTON — The Calvary United" | "Methodist Church in Lewiston is hosting a free, family-friendly service on Sunday, Sept. 25, from 10 a.m. to noon.\n\nThe service will be held at the" |
| 6 | "LEWISTON — The Calvary United" | "Methodist Church in Lewiston is hosting a free community service event on Saturday, May 14, from 10 a.m. to noon.\n\nThe event will be held at the church, located" |
| 4 | "LEWISTON — The Calvary United" | "Church of Christ will be hosting a "Prayer Service" at 6:30 p.m. on Saturday, May 18, at the Calvary United Church of Christ," |

Figure 16: Text completion test cases on quantized OPT-350M.

Table 13: Detection and segmentation performances under different bit widths on COCO2017 *val* dataset. $\dagger$ means models with multi-scaled training. MaskRCNN-SWIN-T$^\dagger$ is evaluated.

| Method | Bit Width | Model Size (MB) | Bbox mAP | | | Mask mAP | | |
|---|---|---|---|---|---|---|---|---|
| | | | mAP | mAP@50 | mAP@75 | mAP | mAP@50 | mAP@75 |
| Original | 32 | 182.59 | 42.7 | 65.2 | 46.8 | 39.3 | 62.2 | 42.2 |
| BLOB-Q | 6 (MP) | 34.24 | 41.8 | 64.4 | 46.5 | 38.9 | 61.5 | 41.9 |
| BLOB-Q | 5 (MP) | 29.53 | 39.6 | 63.1 | 43.0 | 37.3 | 60.1 | 39.6 |
| BLOB-Q | 4 (MP) | 22.82 | 32.9 | 56.0 | 35.3 | 33.2 | 53.8 | 35.4 |

Table 14: Detection and segmentation performances under different bit widths on COCO2017 *val* dataset. $\dagger$ means models with multi-scaled training. MaskRCNN-SWIN-T is evaluated.

| Method | Bit Width | Model Size (MB) | Bbox mAP | | | Mask mAP | | |
|---|---|---|---|---|---|---|---|---|
| | | | mAP | mAP@50 | mAP@75 | mAP | mAP@50 | mAP@75 |
| Original | 32 | 182.59 | 46.0 | 68.1 | 50.4 | 41.6 | 65.2 | 44.7 |
| BLOB-Q | 6 (MP) | 34.24 | 45.0 | 67.4 | 49.3 | 41.1 | 64.6 | 44.2 |
| BLOB-Q | 5 (MP) | 29.53 | 42.5 | 64.9 | 47.2 | 39.9 | 62.3 | 43.4 |
| BLOB-Q | 4 (MP) | 22.82 | 26.2 | 46.8 | 26.5 | 27.7 | 45.8 | 29.5 |

Table 15: Detection and segmentation performances under different bit widths on COCO2017 *val* dataset. $\dagger$ means models with multi-scaled training. MaskRCNN-SWIN-S is evaluated.

| Method | Bit Width | Model Size (MB) | Bbox mAP | | | Mask mAP | | |
|---|---|---|---|---|---|---|---|---|
| | | | mAP | mAP@50 | mAP@75 | mAP | mAP@50 | mAP@75 |
| Original | 32 | 264.16 | 48.2 | 69.9 | 52.8 | 43.2 | 67.1 | 46.0 |
| BLOB-Q | 6 (MP) | 49.53 | 47.2 | 69.4 | 51.8 | 42.7 | 66.6 | 46.0 |
| BLOB-Q | 5 (MP) | 41.28 | 46.0 | 67.9 | 51.0 | 42.0 | 65.2 | 45.3 |
| BLOB-Q | 4 (MP) | 33.02 | 27.9 | 45.4 | 31.0 | 28.4 | 44.4 | 31.3 |

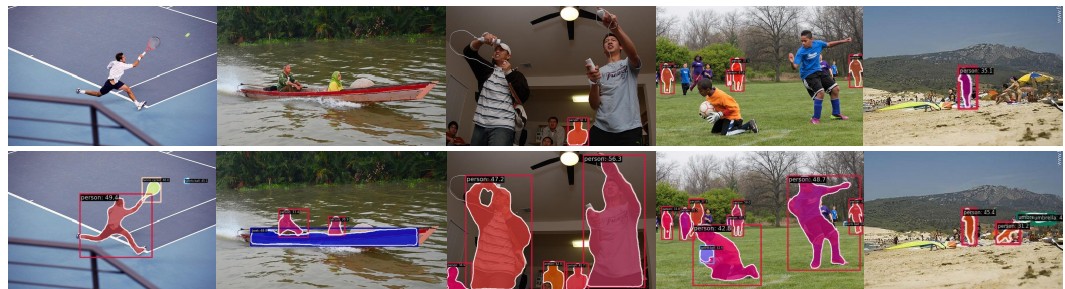

Figure 17: Visualization of the detection results. First row shows uniform INT4 quantizaiton results and second row shows F-LBQ INT4 results.

**Qualitative Results.** Fig. 17 shows the qualitative detection results on COCO 2017 validation images. At bit rates as low as 4-bit, we observe that although there are some cases of incomplete bounding boxes (e.g., the boat in the second image), the detection quality is generally decent and drastically better than the baseline results.

## E  QUANTIZATION WITH MIXED-PRECISION ON HARDWARE

In practice, there are three stages of data movement for Mixed-Precision Quantization on hardware: global memory stage, buffer stage, and processing stage (or compute unit stage). At global memory stage, weights are stored and packed based on bit-width to maximize storage efficiency. Then at buffer stage, weights will be unpacked and aligned before sending to compute unit. For example, if a) memory has a 16-bit word, b) weights are quantized to 4-bit, and c) 8-bit int is used in compute logic, then a 16-bit word will be unpacked to 4 weights and each weight will be padded to a 8-bit integer with zero-padding. Last, the unpacked and padded weights will be sent to compute units. The hardware dynamically processes weights with different bit widths based on the memory packing and alignment mechanisms.

Here is an example when weights are quantized to 4-bit, with bitwise operations involved in the process. Assume the length of a memory word is 16-bit. 4 weights are packed into a single word. Let w1 = 0xA (1010 in binary), w2 = 0x4 (0100 in binary), w3 = 0x5 (0101 in binary), and w4 = 0x7 (0111 in binary) denote 4 weights. They will be packed into a word:

$$packed\_word = [w1\,w2\,w3\,w4] = 1010\,0100\,0101\,0111\ (packed\ into\ a\ 16\ bit\ word).$$

When the packed data is fetched from global memory to buffer, bitwise operations are used to extract the individual values from the packed word. For example, to extract the 4-bit weight w1, perform a bitwise AND operation with a mask and shift the result: mask = 0xF000 (in binary: 1111 0000 0000 0000),

$$extract\ w1: \ packed\_word\ \&\ mask\ >>\ 12.$$

If 8-bit int is used in compute logic, each weight will be padded to a 8-bit integer,

$$padding: \ w1 = 0000\,1010, \ w2 = 0000\,0100, \ w3 = 0000\,0101, \ w4 = 0000\,0111.$$

The padded weights will be then sent to compute unit.

We implemented 8-bit and 4-bit matrix multiplication and convolution for GPU acceleration based on Nvidia's CUTLASS template library[2] as CUDA extensions to python-based main process, and replaced them with Pytorch's default convolution and linear module and tested the speed up on GPUs. Since Nvidia's TensorCore is supported on Ampere machines with the lasted versions of the CUDA compiler, our implementation leverages the TensorOp GEMMs provided by CUTLASS. For INT8 and INT4 convolution, we adopt the implicit convolution for signed 8-bit (`s8`) and signed 4-bit (`s4`) respectively. For INT4 operations, since Pytorch does not support `s4` data type, we first perform packing operations for both inputs and weights (quantized to effective 4-bit but contained in `torch.int8` data type), which packs two adjacent `s4` values into a byte, then transfer them from pytorch in `s8` type to C++. The packed `s8` data are further casted into `s4` type

---

[2]https://github.com/NVIDIA/cutlass

using `reinterpret_cast<cutlass:int4b_t *>`. We implemented the packing for two `s4` values $f_1, f_2$ as below:

$$\text{pack}(f_1, f_2) = f_1 \ll 4 + f_2 + 16 \cdot \mathbb{I}(f_2 < 0). \tag{25}$$

This packing process is implemented in parallel to reduce the overhead when applied to tensors.

To efficiently utilize CUDA kernels, CUTLASS adopts 128-bit vector memory access. As a result, INT8 convolution and matmul require the input channels to be a multiple of 16, and the INT4 ones require a multiple of 32 instead. Therefore, we pad the input channels to meet this alignment requirement for INT4 or INT8 inference. We then utilize the implemented INT4 and INT8 operations to perform mixed-precision quantization on Vision Transformers (ViTs). We constrain the quantization parameters and bit allocation options to only 4 and 8 bits. We evaluate such constrained mixed-precision on DeiT-B when quantied to 6 bits on average. Tab. 16 shows the performance of the inference time on DeiT-B and the speedup under different batch sizes. The Mixed-Precision Quantization at 6 bits on average with the above implementation can bring up to $1.6\times$ speedups.

Table 16: Benchmark performances of Mixed-Precision Quantization at 6 bits on DeiT-B inference using INT8 and INT4 GEMM on Nvidia A100.

| Batch Size | Time Cost (ms) | | | |
|---|---|---|---|---|
| | 32 | 64 | 128 | 256 |
| FP32 | 2.38 | 2.42 | 2.40 | 2.38 |
| 6 Bits | 1.45 | 1.44 | 1.52 | 1.54 |
| Speedup | 1.64 | 1.68 | 1.58 | 1.55 |

### E.1 DISCUSSION ON MEMORY BANDWIDTH

To explore the empirical benifit of adopting a mixed-precisioned quantization recipe compared to uniform quantization, we conducted further comparison of our mixed-precision approach with uniform quantization method, regarding the hardware performance. Firstly, we compare the inference latency with the SOTA uniform quantization method (RepQ-ViT) on an 80GB-A100 GPU platform, with a high memory bandwidth at 2.039 TB/s. Tab. 17 summarizes the inference speed resulting from our mixed-precision quantization approach. In Tab. 17, we choose the settings of two models for both approaches where accuracy loss is less than 0.5%. For all settings, we show the average latency on 8192 test images with batchsize of 64. For DeiT-S, RepQ-ViT (Uniform quantize method) can quantize down to 8-bit to maintain the accuracy at 79.67%, while our method can quantize down to 6-bit with the same level of the accuracy at 79.73% with consistent gain in latency on multiple hardware platforms. We observe the similar gain for DeiT-B. The results show we gain latency speedups from uniform quantization up to 19%, without hurting accuracy.

Table 17: GPU latency (ms) of mixed-precision inference compared to uniform quantization.

| Method | DeiT-S | | DeiT-B | |
|---|---|---|---|---|
| | Top-1 Acc. | Latency (ms) | Top-1 Acc. | Latency (ms) |
| FP | 79.87% | 65.4 | 81.85% | 179 |
| RepQ-ViT (Uniform) | 79.67% (-0.2) | 10.5 | 81.73% (-0.12) | 113 |
| Ours (MP) | **79.73% (-0.14)** | **8.78** | **81.85% (+0)** | **102** |

### E.2 DISCUSSION ON THE EFFECT ON MEMORY BANDWIDTH

We also evaluated the hardware performance on two hardware platforms: TPU V3 and Eyeriss, which are the neural-network-target hardware accelerators targeted for edge devices. The memory bandwidths on these platforms are much less than A100 GPU, at 12.5GB/s and 1GB/s for TPU and Eyeriss respectively. Using the standard SCALE-sim simulator, we simulated the inference speed on DeiT-S and DeiT-B, when deployed onto the two hardware platforms. Tab. 18 summarizes the inference speed resulting from our mixed-precision quantization approach, compared with RepQ-ViT. In Tab. 18, we also choose the settings of both approaches where accuracy loss is less than 0.5%.

As we can see, our mixed-precision quantization approach can further reduce the inference time noticeably by up to 25%, compared with the uniform quantization approach. The improvement comes from the low-bit quantization of our approach. As the proposed mixed-precision quantization can quantize models to lower bit widths, it thus leads to smaller model size and less memory read/write time.

Table 18: Hardware performance on Edge devices with different memory bandwidths.

| Method | DeiT-S | | | DeiT-B | | |
| --- | --- | --- | --- | --- | --- | --- |
| | Top-1 Acc. | Latency on TPU V3 (ms) | Latency on Eyeriss (ms) | Top-1 Acc. | Latency on TPU V3 (ms) | Latency on Eyeriss (ms) |
| FP | 79.87% | 10.8 | 646.4 | 81.85% | 18.6 | 1113.2 |
| RepQ-ViT (Uniform) | 79.67% (-0.2) | 9.1 | 449.7 | 81.73% (-0.12) | 15.7 | 774.4 |
| Ours (MP) | **79.73% (-0.14)** | **6.8** | **337.3** | **81.85% (+0)** | **12.9** | **580.8** |

