# OpenReview forum: "BLOB-Q: Boosting Low Bit ViT Quantization via Global Optimization on Model Distortion"
_ICLR.cc/2026/Conference — ICLR 2026 Conference Withdrawn Submission_

### Official Review · Reviewer_HeS4 · 2025-10-26

**Soundness:** 2
**Presentation:** 3
**Contribution:** 2
**Rating:** 4
**Confidence:** 4

**Summary:**

The paper proposes a Mixed-Precision Post-Training Quantization (PTQ) framework for Vision Transformers (ViTs). Unlike previous approaches that minimize layer-wise reconstruction error, the method directly minimizes the final-layer output error (i.e., model-level distortion). The authors formulate the problem as a global optimization task and introduce a dynamic programming algorithm that finds the globally optimal bit allocation in linear time. Experimental results demonstrate consistent improvements over baseline methods.

**Strengths:**

The formulation based on second-order approximation and the additivity property provides a sound theoretical foundation for global distortion minimization.
The proposed dynamic programming solver is computationally efficient, achieving linear scaling and outperforming prior heuristic search methods.
Theoretical analysis is thorough and well-supported by derivations in both the main text and the supplementary material.
The overall framework is efficient and well-motivated.

**Weaknesses:**

The comparative analysis is limited. Most reported baselines use uniform-bit quantization, making the improvement less surprising. The few mixed-precision baselines included are outdated and insufficient to establish competitiveness against recent methods.

The fine-grained bit allocation across both weights and activations (as shown in Figures 7–8) may introduce nontrivial implementation overhead and complicate hardware deployment compared to uniform-bit schemes.

The assumption of inter-layer independence—and particularly the independence between activation and weight errors—is questionable. Figure 3 implies negligible inter-layer dependency, which seems unrealistic, especially within transformer blocks where correlations between layers of same block are expected. The figure also omits quantitative measures of dependency magnitude. It remains unclear whether the proposed method would remain effective without this assumption.

**Questions:**

Please see Weakness

---

> ### Author Response · Authors · 2025-11-22
>
> > ### (W1) The comparative analysis is limited. Most reported baselines use uniform-bit quantization, making the improvement less surprising. The few mixed-precision baselines included are outdated and insufficient to establish competitiveness against recent methods.
>
> Thanks for the constructive suggestion. During rebuttal, we have included multiple more recent mixed-precision approaches as baselines, and we compare with them on as many models and bit-widths configurations as possible. Specifically, we added 3 more recent mixed-precision methods in our comparison, including PMQ [1 (IJCNN' 23), P$^2$-ViT [2] (VLSI’ 24), and PTMQ [3] (AAAI' 24), under 4-bit, 6-bit and 8-bit on 4 ViT models. As shown in Table R4.1, we can observe that our proposed method maintains it advantages with the newer methods coming in. Our method outperforms other methods by 3.08%, 4.45% and 13.4% for 4/6/8-bit respectively on ViT-S. We observe similar improvements on other three models.
>
> We will accordingly amend the updated comparisons in the revised manuscript. We hope this address your potential concern on the lack in comparisons, and please let us know if any other baselines are missing.
>
>
> Table R4.1 Comparison with more recent Mixed-precision methods.
>
> | Method | Wbit | Abit | ViT-S | ViT-B | DeiT-S | DeiT-B |
> |-----------------|------|------|-------|-------|--------|--------|
> | P$^2$-ViT | 4 MP | 8 | 64.24 | 79.93 | 75.26 | 79.37 |
> | BLOB-Q (Ours) | 4 MP | 4 MP | **67.32** | **79.97** | **77.42** | **79.48** |
> | PMQ [1] | 6 MP | 6 MP | - | 73.33 | 76.68 | 79.64 |
> | P$^2$-ViT [2] | 6 MP | 8 | 32.05 | 82.1 | 77.56 | 80.59 |
> | PTMQ [3] | 6 MP | 6 MP | 76.09 | 77.7 | 78.74 | 80.81 |
> | BLOB-Q (Ours) | 6 MP | 6 MP | **80.53** | **83.66** | **79.53** | **81.58** |
> | P$^2$-ViT | 8 MP | 8 MP |68.14 | 83.00 | 78.41 | 80.93 |
> | Ours MP | 8 MP |8 MP | **81.54** | **84.89** | **81.25** | **81.75** |
>
>
>
> > ### (W2) The fine-grained bit allocation across both weights and activations (as shown in Figures 7–8) may introduce nontrivial implementation overhead and complicate hardware deployment compared to uniform-bit schemes.
>
>
> Thanks for this insight. The implementation overheads during hardware deployment indeed exist. However, the improvements in hardware performance of our quantization method can be mainly attributed to significantly improved model accuracy under lower model sizes. Compared to uniform-bit schemes, our proposed method is able to quantize the models to lower sizes at the same level of model accuracy, which will drastically decrease hardware overhead. Such overhead reduction introduced by mode size reduction is significant enough to offset the overhead caused by the mixed-precision implementation. In Table 14 and Table 15 in the appendix, we compare the performances at the lowest compressed rate for uniform and mixed-precision approaches such that the model accuracy drops are both less than 0.5% from full-precision models.
>
> We provide more details of the Table 14 and Table 15 in the appendix as shown in Table R4.2, Table R4.3. For DeiT-S, RepQ-ViT (Uniform quantize method) can quantize down to 8-bit to maintain the accuracy at 79.67%, while our method can quantize down to 6-bit with the same level of the accuracy with consistent gain in latency on multiple hardware platforms. We observe the similar gain for DeiT-B. This shows that the gain in hardware performance indeed mainly come from lower model size achieved by our method. Thanks again for this valuable question. We will also include the relevant discussion in the revised manuscript.
>
> Table R4.2 Quantized Performances of DeiT-S on various hardware platforms at the critical bit-widths with $<0.5%$ accuracy drops.
> |Method | Bit | Top-1 Accuracy (%) | Latency on GPU (ms) | Latency on TPU V3 (ms) | Latency on Eyeriss (ms) |
> |---|---|--|--|--|--|
> |FP |32|79.87% |65.4 | 10.8|  646.4 |
> |RepQ-ViT |8|79.67% (-0.2) | 10.5| 9.1 | 449.7|
> |Ours | 6| 79.73% (-0.14) | **8.78**| **6.8** | **337.3**|
>
> Table R4.3 Quantized Performances of DeiT-B on various hardware platforms at the critical bit-widths with $<0.5%$ accuracy drops.
> |Method | Bit | Top-1 Accuracy (%) | Latency on GPU (ms) | Latency on TPU V3 (ms) | Latency on Eyeriss (ms) |
> |---|---|--|--|--|--|
> |FP |32|81.85% |179|  18.6|  1113.2 |
> |RepQ-ViT |8|81.73% (-0.12) | 113| 15.7 |  774.4|
> |Ours | 6| 81.85% (+0) | **102**| **12.9**| **580.8** |

---

> > ### Author Response · Authors · 2025-11-22
> > **[Part1/2] Response to Weakness 3**
> >
> > > ### (W3) The assumption of inter-layer independence—and particularly the independence between activation and weight errors—is questionable. Figure 3 implies negligible inter-layer dependency, which seems unrealistic, especially within transformer blocks where correlations between layers of same block are expected. The figure also omits quantitative measures of dependency magnitude. It remains unclear whether the proposed method would remain effective without this assumption.
> >
> > Thank you for all the important questions. We address each of your question one by one in details as below.
> >
> > - **Dependency between Layers of Same Block**:
> >
> > Regarding the concern about the counterintuitive independency between within-block layers especially between weight and activation errors, we provide explanations from the probability perspective. We want to highlight that the introduced independence assumption is regarding to the quantization errors when we quantize weights or activations from the pretrained model, rather than the weight/activation themselves. Although it is intuitive to consider that weights and activations within the same layer should display some interaction, it does not guarantee that the respective quantization error also shares the same intuition. During the calibration process of our post-training quantization on a fixed pretrained model, both weight $\mathbf{W}_l$ and activation $\mathbf{A}_l$ of the same layer are fixed as constants. Furthermore, since the quantized weight $q_1(\mathbf{W}_l)$ and activation $q_2(\mathbf{A}_l)$ are separately obtained with different quantization parameters (e.g. bit-widths, scaling factors, rounding, etc.), the quantization errors $\Delta\mathbf{W}_l = \mathbf{W}_l - q_1(\mathbf{W}_l)$ and $\Delta\mathbf{A}_l = \mathbf{A}_l - q_2(\mathbf{A}_l)$ are also fully generated from different mechanisms without considering each other. Therefore in the post-training setting, the distributions of quantization errors of weights and activations are purely populated by altering quantization parameters and quantization functions, thus can be considered uncorrelated with each other. This is also the same case between different layers within the same block.
> >
> > We also provide empirical studies showing the correlation coefficients and the covariance behaviors for the quantization errors between the same layer weight and activation. As shown in Table R4.4, we observe that the correlation coefficients (0-1 score, 0=fully-uncorrelated, 1=fully correlated) shows low magnitude (up to $10^{-3}$) for all the layers, models and bit-widths, indicating a negligible correlation between weights and activations in ViTs. In terms of covariance behaviors, the average values on off-diagonal positions of the covariance matrices the are also close to zero, with at most $10^{-6}$ magnitude for all the weight-activation pairs in different configurations. This shows the weights and activations from same layer satisfies the independence assumption on the quantization error.
> >
> > Table R4.4 Correlation and covariance statistics of quantization error on same layer weight and activation.
> > | Model  | Bit-width | Layer | Correlation | Avg. On-diagonal Covariance | Avg. Off-diagonal Covariance |
> > |---|---|--|---|---|--|
> > | ViT-S   | 6 | blks.4.attn.proj  | $1.78 \times 10^{-3}$   | $8.16 \times 10^{-5}$      | $5.64 \times 10^{-8}$ |
> > | ViT-S   | 6 | blks.0.attn.qkv  | $2.18 \times 10^{-3}$   | $6.74 \times 10^{-4}$      | $2.40 \times 10^{-7}$  |
> > | ViT-S   | 6  | blks.9.mlp.fc1 | $2.21 \times 10^{-3}$   | $4.53 \times 10^{-4}$      | $1.35 \times 10^{-7}$  |
> > | Swin-S  | 2  | layers.2.blks.6.attn.proj | $3.89 \times 10^{-4}$   | $8.24 \times 10^{-3}$      | $9.44 \times 10^{-7}$   |
> > | Swin-S  | 2  | layers.2.blks.11.mlp.fc2  | $2.31 \times 10^{-3}$   | $5.34 \times 10^{-2}$      | $1.87 \times 10^{-6}$ |
> > | Swin-S  | 2   | layers.2.blks.3.mlp.fc2   | $6.20 \times 10^{-4}$   | $8.35 \times 10^{-3}$      | $1.52 \times 10^{-6}$  |
> >
> >
> > - **Missing Quantitative Scales in Fig. 3**:
> >
> > We have added the missing quantitative scales in the Fig. 3 in the appendix of the revised manuscript. The darkest color in Fig. 3 corresponds to $E(||\Delta\mathbf{W}_i \Delta\mathbf{W}_j||^2)=1$, and the white color corresponds to $0$. We measure the quantitative statistics of Fig.3 in the Table R4.5, including the colormap ranges, and the average values of the covariance map on on-diagonal and off-diagonal positions respectively. For all the models, we observe a distinct difference between on-diagonal and off-diagonal covariances by $10^4$ in magnitude, which is large enough to show that the quantization errors between different layers are indeed independent from each other.
> >
> > Thanks again for the good question. We will include the above discussions in the revised manuscript.

---

> > > ### Author Response · Authors · 2025-11-22
> > > **[Part2/2] Response to Weakness 3**
> > >
> > > Table R4.5 Quantitative statistics of Fig.3 in the appendix.
> > >
> > > |Model |ViT-S| ViT-B |DeiT-S |DeiT-B|
> > > |---|---|--|---|--|
> > > |Value Range | $0\sim 1.27\times 10^{-4}$ | $0\sim 4.43\times 10^{-5}$ | $0\sim 7.98\times 10^{-5}$ | $0\sim 2.07\times 10^{-4}$ |
> > > |On-diagonal | $5.27\times 10^{-5}$ |$1.10\times 10^{-5}$ | $3.87\times 10^{-5}$ |$3.91\times 10^{-5}$ |
> > > |Off-diagonal | $4.02\times 10^{-9}$ |$1.98\times 10^{-9}$ | $7.47\times 10^{-9}$ |$4.28\times 10^{-9}$ |
> > >
> > >
> > > **References**
> > >
> > > [1] Patch-wise mixed-precision quantization of vision transformer. IJCNN '23.
> > >
> > > [2] P^2-ViT: Power-of-Two Post-Training Quantization and Acceleration for Fully Quantized Vision Transformer. VLSI' 24.
> > >
> > > [3] Post-training multi-bit quantization of neural networks. AAAI' 24.

---

> ### Author Response · Authors · 2025-11-24
> **Please kindly check our responses and revisions**
>
> Dear Reviewer HeS4,
>
> \
> Thank you again for your careful and extremely insightful comments and many constructive suggestions.
>
> \
> In our previous responses, we have carefully provided detailed explanations and additional analysis to your comments one-by-one. We have also accordingly revised the relevant places in the manuscript.
>
> \
> It would be very appreciated if you can take a look, and please let us know if there are any remaining concerns.
>
> \
> Thanks a lot for your time and effort for the review.
>
> \
> Best regards,
>
> Authors

---

> > ### Comment · Reviewer_HeS4 · 2025-11-26
> >
> > I have read the rebuttal, as far as I know, OMPQ/EMQ only use mixed-precision quantization for model weight, while BLOB-Q use mixed-precision for both weight and activation. What is the performance of BLOB-Q when using mixed-precision for only model weight (keeping bit of activation fixed same as EMQ/OMPQ), compared with other methods? Same question for other mixed-precision  baselines that the author provide.

---

> > > ### Author Response · Authors · 2025-11-27
> > >
> > > > ### I have read the rebuttal, as far as I know, OMPQ/EMQ only use mixed-precision quantization for model weight, while BLOB-Q use mixed-precision for both weight and activation. What is the performance of BLOB-Q when using mixed-precision for only model weight (keeping bit of activation fixed same as EMQ/OMPQ), compared with other methods? Same question for other mixed-precision baselines that the author provide.
> > >
> > > Thanks for your time in rebuttal and responsible review. Following your suggestion, we have conducted additional studies when only allocating bit-widths for weights using our proposed BLOB-Q, while keeping the activation bit-widths uniformly as the same as EMQ/OMPQ. Then, we compare the performance of BLOB-Q with EMQ and OMPQ methods when we all perform mixed-precision bit-allocation only on weights. Specifically, we fixed the activation bit-widths to the same as the target overall bit-widths of weight, e.g. W-(4 MP)/A4, W-(6 MP)/A6 and W-(8 MP)/A8. As for other mixed-precision baselines, we found that P$^2$-ViT already fixes activation bit-widths uniformly. Other mixed-precision baselines who published their codes, including Percentile, PMQ and PTMQ, we evaluate their results with uniform activation bit-widths.
> > >
> > > As shown in Table R4.5, we first observe that the proposed BLOB-Q with uniform-precision activations still display consistent performance improvements from EMQ and OMPQ. Specifically, it outperforms EMQ and OMPQ by at least 8.31%, 9.51%, 30.95% and 9.47% for 4 ViT models respectively on 4-bit. We observe similar improvements on other bit-widths. We have also evaluated Percentile, PMQ and PTMQ and P$^2$-ViT with uniform-precision activations, and our results consistently outperforms theirs. We notice that our method received slight accuracy drops under uniform-precision activations. This indicates that including activations into the bit-allocation framework is indeed an effective strategy, and our approach provides an optimal bit-allocation for both weights and activations.
> > >
> > > Table R4.5 Additional comparisons with EMQ and OMPQ under mixed-precision weights and uniform-precision activations.
> > > |Method | W-bit | A-bit | ViT-S| ViT-B | DeiT-S | DeiT-B|
> > > |---|---|--|--|--|--|--|
> > > |EMQ|4 MP| 4 |57.18| 70.43|44.97|69.93|
> > > |OMPQ|4 MP| 4 | 40.75|  66.41|41.36| 67.28|
> > > |P$^2$-ViT|4 MP|4| 30.54 | 65.79 | 39.89 | 65.74 |
> > > |BLOB-Q (Ours fixed-bit activation)| 4 MP | 4| 65.49 | 79.94 | 75.92 | 79.40 |
> > > |BLOB-Q (Ours)| 4 MP | 4 MP | **67.32** | **79.97** | **77.42** | **80.30** |
> > > |EMQ|6 MP| 6| 79.54|82.89|73.69|78.48|
> > > |OMPQ|6 MP| 6| 78.46| 82.58|73.46|77.98|
> > > |Percentile (fixed-bit activation)|6 MP|6 |62.81 | 72.89 | 69.84 | 71.82 |
> > > |PMQ (fixed-bit activation)|6 MP|6|29.67 | 72.9 | 76.47 | 79.12 |
> > > | P$^2$-ViT | 6 MP | 6 | 30.68 | 79.68 | 77.21 | 80.1 |
> > > | PTMQ (fixed-bit activation) | 6 MP | 6| 75.25 | 76.94 | 77.85 | 79.86 |
> > > | BLOB-Q (Ours fixed-bit activation)| 6 MP |6| 79.69 | 83.02 | 78.41 | 80.64 |
> > > | BLOB-Q (Ours)| 6 MP |6 MP| **80.73** | **84.23** | **79.53** | **81.85** |
> > > |EMQ |8 MP| 8| 81.39 | 84.82 | 76.25 |80.36|
> > > |OMPQ|8 MP| 8|81.37|84.77|76.11|80.26|
> > > |Percentile (fixed-bit activation)|8 MP|8 |78.41 | 79.97|71.22 | 73.89 |
> > > | P$^2$-ViT | 8 MP | 8 |68.14 | 83.00 | 78.41 | 80.93 |
> > > |BLOB-Q (Ours fixed-bit activation)| 8 MP | 8| 81.41 | 84.86 | 78.93 | 81.04 |
> > > |BLOB-Q (Ours)| 8 MP | 8 MP|**81.54** | **84.89** | **81.25** | **81.75** |
> > >
> > >
> > > Thanks again for all your very valuable comments. We hope our responses clarify your concerns and questions. Please also let us know if you have further concerns, and we will be glad to continue to address them.
> > >
> > > **References:**
> > >
> > > [4] "Post-training quantization for vision transformer."  NeurIPS'21.

---

### Official Review · Reviewer_JeSs · 2025-10-29

**Soundness:** 3
**Presentation:** 3
**Contribution:** 3
**Rating:** 8
**Confidence:** 3

**Summary:**

The paper proposes output-based model distortion as a metric of layerwise quantization impact on the output and additivity where the output distortion is a sum of layerwise quantization impacts.
Based on this, the paper presents a dynamic programming based solution to determine per-layer precision while meeting the given bit constraint.

**Strengths:**

Compared with most of quantization methods which focus on layerwise distortion minimization or training loss minimization, the presented idea of output distortion minimization looks effective according to experimental results.
Additivity property, which I think is not new but very well studied in detail in the paper, is exploited in dynamic programming to explore the entire space of bit allocation.
Ablation studies (in the main paper and appendices) are quite extensive.

**Weaknesses:**

Appendix D.6 reports a preliminary experiment on language model.
It would be nice if more detailed and quantitative analysis were provided.

**Questions:**

The proposed method looks quite effective on ViTs.
How would it be applied to language models?
In other words, how can we address the basic assumptions of the proposed method, zero mean, inter-layer independence, linearity (to ignore higher order terms), ...?

---

> ### Author Response · Authors · 2025-11-23
> **[Part 1/2] Response to comments**
>
> > ### (W1) Appendix D.6 reports a preliminary experiment on language model. It would be nice if more detailed and quantitative analysis were provided.
>
> Thanks for this constructive suggestion. We have provided more comprehensive results on language models, including quantization results on more language models as well as results on more datasets of the proposed mixed-precision quantization method. We also provide some text completion case studies to aid the understanding of the performance of our quantized language models.
>
> - **More Dataset Results**:
>
> To evaluate the generalizability of our method on various language tasks, apart from the wikitext2 dataset that we have provided, we have evaluated our method on two more text completion benchmarking datasets, including PTB and C4, which extend the corpus coverage to more topics and language capabilities. As shown in Table R3.1, we show the performances of the quantized OPT-350Mon Wikitext2, PTB and C4 datasets in form of perplexity scores which indicates the accuracy of the token prediction. We observe similar trends on the two added dataset as on Wikitext2, where our quantized models only drop by at most 8.87 in perplexity down to 4-bit from full-precision model. This indicates high preservation of language capability for the proposed model. This further evaluates the generalizability of the proposed method on language model.
>
> - **More Model Results**:
>
> Apart from the extended dataset evaluation, we have also conducted experiments on another two widely used decoder-only language models, BLOOM-560M and Qwen2.5-0.5B. In Table R3.2, we show the performances of newly added language models on 3 datasets as in Table R3.1. We observe the perplexity scores of the quantized models show minimal drops in perplexity down to 6-bit, with up to down to 6.66 and 3.93 for BLOOM-560M and Qwen2.5-0.5B respectively across three different datasets. This shows that our method can be generalized to various language models with different architectures.
>
> - **Text Completion Results**:
>
> To facilitate a more intuitive understanding of our performances on the quantized language models, we also provide text completion case studies and show the input prompt and the completed sentences from the quantized models. We list the input prompts and output responses of the quantized models at 8/6/4-bit as well as the full-precision OPT-350M in Table R3.3. We observe that the text completion quality is mostly maintained in quantized model down to 4-bit, in terms of the grammar and semantical coherencies, which is consistent with the perplexity results as shown in the above study in Table R3.1.
>
> We will include the comprehensive discussions about language model results in the revised manuscript. Please also let us know if there is any other results you want to check and we will continue to conduct additional studies.
>
> Table R3.1 More dataset results of OPT-350M model.
> | Model | Method | Bit-width| Wikitext2 | PTB | C4 |
> |----|-----|----|----|---|---|
> | OPT-350M | FP | 32 | 23.5607 |21.0897|21.0934|
> | OPT-350M | BLOB-Q | 8 MP | 25.7557 |21.0882|21.1012|
> | OPT-350M | BLOB-Q | 6 MP | 25.9515 |21.4077|21.3307|
> | OPT-350M | BLOB-Q | 4 MP | 34.6864|29.9613|27.1915|
>
>
> Table R3.2 More perplexity results on more language models.
> | Model | Method | Bit-width| Wikitext2 | PTB | C4 |
> |----|-----|----|----|---|---|
> | BLOOM-560M | FP | 32 | 19.4163 | 33.6544 | 25.1162 |
> | BLOOM-560M | BLOB-Q | 8 MP | 19.4321 | 33.6591 | 25.126 |
> | BLOOM-560M | BLOB-Q | 6 MP | 22.2007 | 40.3169 | 28.2866 |
> | Qwen2.5-0.5B | FP | 32 | 11.5153 | 20.4461 | 19.0765 |
> | Qwen2.5-0.5B | BLOB-Q | 8 MP | 11.5321 | 20.4937 | 19.1084 |
> | Qwen2.5-0.5B | BLOB-Q | 6 MP | 13.5852 | 24.3803 | 22.3268 |
>
> Table R3.3 Text completion test cases of OPT-350M on 3 datasets.
> |Bit| Input | Output |
> |--|--|--|
> | FP | "Robert Boulter is an English" | "writer and journalist. He is the author of the novel The Last of the Mohicans, which was nominated for the Booker Prize in 2013. He is also the author of the novel The Last of the Mohicans" |
> | 8 | "Robert Boulter is an English" | "writer and critic. He is the author of the best-selling novel The Last of the Mohicans, and the forthcoming book The Last of the Mohicans: The Story of the Last of the Mohicans."|
> | 6 |"Robert Boulter is an English" | "writer and critic. He is the author of the novel The Last of the Mohicans, which won the National Book Award for Fiction in 2012. He is the author of the novel The Last of the Mohicans" |
> | 4 | "Robert Boulter is an English" | "-born, British-based, and Canadian-based, and he is a member of the British Association of Professional Engineers (BAPE).\n\nRobert Boulter is a member of the British Association" |

---

> > ### Author Response · Authors · 2025-11-23
> > **[Part 2/2] Response to comments**
> >
> > > ### (Q1) The proposed method looks quite effective on ViTs. How would it be applied to language models? In other words, how can we address the basic assumptions of the proposed method, zero mean, inter-layer independence, linearity (to ignore higher order terms), ...?
> >
> > Thanks for the insightful comment. To evaluate the theoretical generalizability of our method to language task and model, we analyze their compliance to the theoretical components used in our approach, including the zero-mean, inter-layer independency, and additivity property assumptions. We provide detailed quantitative analysis as below.
> >
> > - **Zero-mean assumption**:
> > We provide additional empirical evidence and the applicability of the zero-mean assumption on language models. Similar to Fig. 2 in the appendix, we analyze the distributions of the weight quantization error of various layers in 3 language models, OPT-350M, BLOOM-460M and Qwen2.5-0.5B, when we quantize their layer weights to 6-bit. As shown in Table R3.4, we observe that the quantization error distributions are centered very close to zero, with the distribution centers at the magnitude of $10^{-8}\sim 10^{-6}$. This supports the zero-mean assumption on language models.
> >
> > - **Inter-layer independency assumption**:
> > We further evaluate the empirical evidence of inter-layer quantization error independency assumption on language models. We calculate the covariance matrices of quantization errors of different layers $\mathbf{E}(\Delta\mathbf{W}_i \Delta\mathbf{W}_j)$, and we calculate the average value of the on-diagonal against off-diagnoal positions of covariance matrices, i.e. same layer $i=j$ against different layers ($i\neq j$). As we show in the Table R3.5, we observe that the inter-layer covariances displayed in these language models are at the magnitude of $10^{-5}$, which is drastically smaller than the same-layer variances (on-diagonal values), supporting the inter-layer independency for language models.
> >
> > - **Additivity Property**:
> > We randomly select different pairs of layers in the language models, and then apply quantization to each layers respectively. We then sample the Left-hand-side (LHS) of the additivity property equation (Eq.4), i.e. the real model output distortion when we quantize the two layers at the same time, and also the Right-hand-side (RHS), i.e. the estimated model output distortion when we quantize them separately. We calculate the relative error between LHS and RHS (i.e. $Err=\frac{LHS - RHS}{|LHS|}$), and we plot the distribution of this relative error under different layer pairs. Table R3.6 shows that the relative errors are very close to zero, with $<8.1\%$ of the distortion itself on average. This shows that the additivity property is still applicable to the language models under various bit-widths.
> >
> > Thanks again for the insightful question. We will also add the discussions and the relevant visualized figures in the revised manuscript.
> >
> > Table R3.4 Distribution of quantization error of various layers in language models.
> > | Model | layerID | 25th Percentile  | Median   | 75th Percentile | Mean | Variance |
> > |--|--|--|--|--|--|--|
> > | OPT-350M | blks.12.attn.q | $-9.11 \times 10^{-4}$ | $-2.11 \times 10^{-5}$ | $8.7 \times 10^{-4}$ | $7.78 \times 10^{-7}$ | $1.09 \times 10^{-6}$ |
> > | OPT-350M | blks.18.attn.q | $-9.3 \times 10^{-4}$ | $-1.6 \times 10^{-5}$ | $8.94 \times 10^{-4}$ | $1.53 \times 10^{-6}$ | $1.15 \times 10^{-6}$ |
> > | BLOOM-560M| blks.6.attn.q | $-6.50\times 10^{-4}$ | $-4.30\times 10^{-5}$ | $5.64\times 10^{-4}$| $9.65\times 10^{-8}$ | $7.58\times 10^{-7}$ |
> > | BLOOM-560M|blks.12.attn.q | $-7.1\times 10^{-4}$ | $-1.1\times 10^{-4}$ | $4.9\times 10^{-4}$| $-4.66\times 10^{-7}$ | $7.65\times 10^{-7}$ |
> > | Qwen2.5-0.5B|blks.6.attn.q | $-6.6\times 10^{-4}$ | $-6.2\times 10^{-5}$ | $5.33\times 10^{-4}$| $6.99\times 10^{-7}$ | $7.87\times 10^{-7}$ |
> > | Qwen2.5-0.5B|blks.18.attn.q | $-6.6\times 10^{-4}$ | $-5.9\times 10^{-5}$ | $5.42\times 10^{-4}$| $1.98\times 10^{-6}$ | $8.78\times 10^{-7}$ |
> >
> > Table R3.5 Inter-layer dependency of quantization errors statistics on language models.
> > | Model | OPT-350M | BLOOM-560M | Qwen2.5-0.5B|
> > |---|---|--|---|
> > | On-diagonal|$8.74 \times 10^{0}$|$9.06 \times 10^{0}$ | $2.58\times 10^0$ |
> > | Off-diagonal|$-6.28 \times 10^{-5}$|$3.63 \times 10^{-5}$ | $-4.7\times 10^{-5}$ |
> >
> > Table R3.6 Relative error of Additivity property on language models. We analyze the percentiles and mean variance of the distribution sampling different layer pairs.
> > | Model | Bit | 25th Percentile | Median | 75th Percentile | Mean | Variance |
> > |---|---|---|---|---|---|---|
> > | BLOOM-560M|8|$-0.0046$|$0.002$|$0.0152$|$0.0576$|$0.1868$|
> > | BLOOM-560M|6|$-0.0126$|$0.0037$| $0.0335$|$0.0571$|$0.1412$|
> > | OPT-350M|8|$-0.0039$|$0.0002$|$0.0029$|$-0.0015$|$0.0025$|
> > | OPT-350M|6|$-0.0155$|$-0.0008$|$0.0172$|$-0.0183$|$0.0293$|
> > | Qwen2.5-0.5B|8|$-0.0125$|$-0.0008$|$0.0085$|$-0.0813$|$0.3275$|
> > | Qwen2.5-0.5B|6|$-0.0306$|$0.0032$|$0.0571$|$0.0349$|$0.1064$|

---

> > > ### Comment · Reviewer_JeSs · 2025-11-25
> > >
> > > The authors well addressed my concerns. Thank you!

---

> > > > ### Author Response · Authors · 2025-11-27
> > > >
> > > > Dear Reviewer JeSs, we thank you again for your careful review and valuable comments.

---

### Official Review · Reviewer_qybZ · 2025-10-30

**Soundness:** 2
**Presentation:** 3
**Contribution:** 2
**Rating:** 4
**Confidence:** 3

**Summary:**

This paper introduces a Mixed-Precision Post-Training Quantization (PTQ) framework for Vision Transformers that minimizes final-layer output error rather than layer-wise reconstruction error. The authors formulate bit allocation as a global optimization problem and develop a dynamic programming algorithm that finds the optimal solution in linear time.

**Strengths:**

- The paper is well-written
- Theoretical justification.
- The overall framework is efficient and well-motivated.

**Weaknesses:**

- Figure 3 in the appendix only indicates that the dependency of quantization error across layers is smaller than the squared quantization error of each layer. However, it is not clear whether the product of quantization errors between any two layers is significantly smaller or approximates zero. The fact that layers are quantized independently does not intuitively imply that the product of errors equals zero.

**Questions:**

- Tables 14 and 15 demonstrate that the mixed-precision approach is faster and consumes less memory than the uniform-bit approach. How is this possible? What is the reason behind this?

- Is it possible to replace the bit-width allocation of stronger mixed-precision methods (e.g., EMQ/OMPQ/...) to evaluate how much stronger the proposed mixed-precision approach is?

---

> ### Author Response · Authors · 2025-11-23
> **Response to Weakness 1**
>
> > ### (W1) Figure 3 in the appendix only indicates that the dependency of quantization error across layers is smaller than the squared quantization error of each layer. However, it is not clear whether the product of quantization errors between any two layers is significantly smaller or approximates zero. The fact that layers are quantized independently does not intuitively imply that the product of errors equals zero.
>
> Thank you for the very insightful question. To further address the concern whether if the products of quantization errors of different layers are also close to zero, i.e. $E(\Delta\mathbf{W}_i \Delta\mathbf{W}_j)\rightarrow 0 , \forall i\neq j$, we conducted additional evaluation on the ViT models, and we provide detailed quantitative statistics as below. We list the values of the product of quantization errors $E(\Delta\mathbf{W}_i \Delta\mathbf{W}_j)$ of different layers when quantizing to different bit-widths, containing all the layer pairs for ViT models as in Table R2.1. We observe that the magnitude of the product of the quantization errors between two different layers are very low. Their average magnitudes remain below approximately $10^{-5}$, $10^{-3}$ and $10^{-3}$ for 8/6/4-bit respectively, indicating that their correlations are negligible. We have added this additional discussion in the revised manuscript.
>
> Then, we provide explanations to aid the intuitive understanding of this phenomenon. Given a fixed pretrained model, when we consider the impact of applying different quantization strategies such as bit-widths the original model, the weights of the different layers $\mathbf{W}_i$ and activation $\mathbf{W}_j$ can be regarded as constants. Furthermore, since the quantized weights $q_1(\mathbf{W}_i)$ and $q_2(\mathbf{W}_j)$ are separately obtained with different quantization parameters (e.g. bit-widths, scaling factors, rounding, etc.), the quantization errors $\Delta\mathbf{W}_i = \mathbf{W}_i - q_1(\mathbf{W}_i)$ and $\Delta\mathbf{W}_j = \mathbf{W}_j - q_2(\mathbf{W}_j)$ are also fully generated from different mechanisms without considering each other. Therefore in the post-training setting, the distributions of quantization errors of weights in different layers are purely populated by altering quantization parameters and quantization functions, thus can be considered uncorrelated with each other.
>
> Table R2.1 Statistics of product of quantization errors between layers $E(\Delta\mathbf{W}_i \Delta\mathbf{W}_j)$. We report the average values for all pairs of different layers.
> | Bit-width | ViT-S| ViT-B | DeiT-S | DeiT-B |
> |---|---|--|----|----|
> | 8-bit | $-6.3 \times 10^{-5}$ | $-4.9 \times 10^{-6}$ | $-1\times 10^{-5}$ | $-3.2 \times 10^{-5}$ |
> | 6-bit | $-4.3 \times 10^{-4}$ | $-1.2 \times 10^{-4}$ | $1.06\times 10^{-3}$ | $2.87\times 10^{-4}$ |
> | 4-bit | $5.70 \times 10^{-4}$ | $1.13 \times 10^{-3}$ | $1.1\times 10^{-3}$ | $2.43 \times 10^{-3}$|

---

> > ### Author Response · Authors · 2025-11-23
> > **Response to Questions**
> >
> > > ### (Q1) Tables 14 and 15 demonstrate that the mixed-precision approach is faster and consumes less memory than the uniform-bit approach. How is this possible? What is the reason behind this?
> >
> >
> > Thanks for the good question. The major reason behind the improvements in latency and memory is that the proposed mixed-precision method can quantize the model to lower model sizes than uniform approaches, under the same level of model accuracy. Lower model size consequently drastically decreases the memory consumption of the model weights and intermediate variables required during inference. This further reduce the memory read and write time as well as the computation required for the inference, leading to lower latency. In the appendix's Table 14 and Table 15, we compared the performances on hardware platforms for uniform and mixed-precision approaches such that the model accuracy is on the same level (accuracy drop <0.5%).
> >
> > We provide more results of hardware performances as shown in Table R2.2, Table R2.3. For DeiT-S, RepQ-ViT (Uniform quantize method) can quantize down to 8-bit to maintain the accuracy at 79.67%, while our method can quantize down to 6-bit with the same level of the accuracy at 79.73%. The smaller model size achieved by our method leads to smaller memory consumption, resulting in lower inference latency and higher hardware performance. We observe the similar gain for DeiT-B. This shows that the gain in hardware performance indeed mainly come from lower model size achieved by our method. Thanks again for this valuable question. We will add the relevant discussion in the revised manuscript.
> >
> > Table R2.2 Quantized Performances of DeiT-S on various hardware platforms at the critical bit-widths with <0.5% accuracy drops.
> > |Method | Bit | Top-1 Accuracy (%) | Latency on GPU (ms) | Latency on TPU V3 (ms) | Latency on Eyeriss (ms) |
> > |---|---|--|--|--|--|
> > |FP |32|79.87% |65.4 | 10.8|  646.4 |
> > |RepQ-ViT |8|79.67% (-0.2) | 10.5| 9.1 | 449.7|
> > |Ours | 6| 79.73% (-0.14) | **8.78**| **6.8** | **337.3**|
> >
> > Table R2.3 Quantized Performances of DeiT-B on various hardware platforms at the critical bit-widths with <0.5% accuracy drops.
> > |Method | Bit | Top-1 Accuracy (%) | Latency on GPU (ms) | Latency on TPU V3 (ms) | Latency on Eyeriss (ms) |
> > |---|---|--|--|--|--|
> > |FP |32|81.85% |179|  18.6|  1113.2 |
> > |RepQ-ViT |8|81.73% (-0.12) | 113| 15.7 |  774.4|
> > |Ours | 6| 81.85% (+0) | **102**| **12.9**| **580.8** |
> >
> > > ### (Q2) Is it possible to replace the bit-width allocation of stronger mixed-precision methods (e.g., EMQ/OMPQ/...) to evaluate how much stronger the proposed mixed-precision approach is?
> >
> > Thanks for the good suggestion. During rebuttal, following your suggestion, we evaluated the performance when integrating the bit-width allocations of 2 recent SOTA mixed-precision methods respectively into our approach, including EMQ [1] and OMPQ [2], and then evaluated their results. Specifically, we replace our original bit-allocation with theirs once at a time, while keeping other parts of our approach the same (e.g. quantizer choices). As shown in Table R2.4, we compare the performances of our method as well as when we combine the other two bit allocation strategies. "EMQ/OMPQ + Our quantizers" denotes the results when we replace the bit-allocation strategies of EMQ/OMPQ respectively with our bit allocation strategy; "Our bit allocation + Our quantizers" denotes our method. We observe that our method outperforms both "EMQ + Our quantizers" and "OMPQ + Our quantizers". Under 4-bit, we observe significant accuracy discrepancies by at least 10.29% and 13.19% from EMQ and OMPQ respectively. Under 6/8-bit, we observe similar accuracy discrepancies between the integration results and our original results. This shows that our bit allocation strategy is indeed stronger than existing bit-allocation strategies.
> >
> > Table R2.4 Integration results with other mixed-precision methods with our quantizers
> > |Method | Bit | ViT-S| ViT-B | DeiT-S | DeiT-B|
> > |---|---|--|--|--|--|
> > |EMQ + Our quantizers|4 MP|57.03| 70.28|44.72|69.71|
> > |OMPQ + Our quantizers|4 MP| 40.04|  66.13|41.17| 67.11|
> > | Our bit allocation + Our quantizers| 4 MP | **67.32** | **79.97** | **77.42** | **80.30** |
> > |EMQ + Our quantizers|6 MP| 79.30|82.78|73.47|78.37|
> > |OMPQ + Our quantizers|6 MP| 78.24| 82.37|73.23|77.91|
> > | Our bit allocation + Our quantizers| 6 MP | **80.73** | **84.23** | **79.53** | **81.85** |
> > |EMQ + Our quantizers|8 MP| 81.36 | 84.86 | 76.08 |80.19|
> > |OMPQ + Our quantizers|8 MP| 81.13|84.74|75.96|80.08|
> > |Our bit allocation + Our quantizers| 8 MP | **81.54** | **84.89** | **81.25** | **81.75** |
> >
> >
> > **Reference**
> >
> > [1] Dong, Peijie, et al. "Emq: Evolving training-free proxies for automated mixed precision quantization."ICCV'23.
> >
> > [2] Ma, Yuexiao, et al. "Ompq: Orthogonal mixed precision quantization." AAAI'23.

---

> ### Author Response · Authors · 2025-11-24
> **Please kindly check our responses and revisions**
>
> Dear Reviewer qybZ,
>
> \
> Thank you again for your careful and extremely insightful comments and many constructive suggestions.
>
> \
> In our previous responses, we have carefully provided detailed explanations and additional analysis to your comments one-by-one. We have also accordingly revised the relevant places in the manuscript.
>
> \
> It would be very appreciated if you can take a look, and please let us know if there are any remaining concerns.
>
> \
> Thanks a lot for your time and effort for the review.
>
> \
> Best regards,
>
> Authors

---

> ### Comment · Reviewer_qybZ · 2025-11-26
>
> Thanks authors for your rebuttal. This addresses most of my concerns. Depending on your answers, could you please clarify the following?
>
> 1. How does the sum of off-diagonal terms ($\sum_{i \neq j} \delta W_i \delta W_j$) compare to the sum of diagonal terms ($\sum_i \delta W_i^2$)?
>
> 2. Is $\delta W_i \delta A_j$ actually insignificant compared to $\delta W_i^2$?

---

> > ### Author Response · Authors · 2025-11-27
> >
> > > ### 1.  How does the sum of off-diagonal terms ($\sum_{i\neq j} \Delta W_i \Delta  W_j$) compare to the sum of diagonal terms ($\sum_{i} \Delta W_i^2$)?
> >
> > Thanks for your follow-up comments. We have conducted additional evaluation to compare the sums of on-diagonal and off-diagonal values of the term $\Delta \mathbf{W}_i \Delta  \mathbf{W}_j$ on all the layer pairs on ViTs, and listed below in Table R2.5.
> > We observe that the sum of on-diagonal terms on all layer is indeed noticeably larger compared to off-diagonal layer pairs, by at least 65x, 59x and 108x times for 8/6/4-bit respectively. This shows that the even though there are much more layer pairs for off-diagonal positions than on-diagonal, the sum of off-diagonal terms are still not much smaller than the on-diagonal ones.
> >
> > Table R2.5 Sum of on-diagnoal v.s. off-diagnoal $\Delta\mathbf{W}_i \Delta\mathbf{W}_j$.
> > | | Bit | ViT-S| ViT-B | DeiT-S | DeiT-B |
> > |---|---|--|----|----|--|
> > Sum of On-diagnoal | 8-bit | $9.69$ | $13.87$ | $8.73$ | $32.32$ |
> > Sum of Off-diagnoal | 8-bit | $-0.15$ | $-0.01$ | $-0.03$ | $-0.07$ |
> > Sum of On-diagnoal | 6-bit | $59.64$ | $68.49$ | $45.66$ | $178.66$ |
> > Sum of Off-diagnoal | 6-bit | $-1.02$ | $-0.27$ | $-0.18$ | $0.67$ |
> > Sum of On-diagnoal | 4-bit | $381.03$ | $319.13$ | $279.89$ | $1131.37$ |
> > Sum of Off-diagnoal | 4-bit | $-1.39$ | $2.76$ | $2.59$ | $5.97$ |
> >
> >
> > > ### 2.  Is $\Delta W_i \Delta  W_j$ actually insignificant compared to $\Delta W_i^2$?
> >
> > To comprehensively compare the on-diagonal and off-diagonal values, we provide further analysis on the ratio between the on-diagonal values against off-diagonal values, i.e. $E(\Delta W_i \Delta  W_j) / E(\Delta W_i^2)$. As listed in Table 2.6, we compared the average values between On-diagonal and Off-diagonal terms, and calculated their ratio, for different ViT models. We notice that the ratio between $\Delta W_i \Delta  W_j$ and $\Delta W_i^2$ on average are consistently distinct enough, at the magnitude of $10^4\sim 10^5$ across all models and bit-widths. This shows that the Off-diagonal terms are indeed insignificant compared to on-diagonal ones.
> >
> > Table R2.6 More statistics of ratio between on-diagonal v.s. off-diagonal $\Delta\mathbf{W}_i \Delta\mathbf{W}_j$.
> > | | Bit | ViT-S| ViT-B | DeiT-S | DeiT-B |
> > |---|---|--|----|----|--|
> > On-diagonal  $E(\Delta W_i^2)$ | 8 | $1.98 \times 10^{-1}$ | $2.83 \times 10^{-1}$ | $1.78\times 10^{-1}$ | $6.59\times 10^{-1}$ |
> > Off-diagonal $E(\Delta W_i \Delta  W_j)$ | 8 | $-6.3 \times 10^{-5}$ | $-4.9 \times 10^{-6}$ | $-1\times 10^{-5}$ | $-3.2 \times 10^{-5}$ |
> > Ratio $\|E(\Delta W_i \Delta  W_j) / E(\Delta W_i^2)\|$ | 8 | $3.18\times 10^{-4}$ | $1.73\times 10^{-5}$ | $5.73\times 10^{-5}$ | $4.86\times 10^{-5}$ |
> > On-diagonal  $E(\Delta W_i^2)$ | 6 | $1.22 \times 10^{0}$ | $1.39 \times 10^{0}$ | $9.32\times 10^{-1}$ | $3.65 \times 10^{0}$ |
> > Off-diagonal $E(\Delta W_i \Delta  W_j)$ | 6 | $-4.3 \times 10^{-4}$ | $-1.2 \times 10^{-4}$ | $1.06\times 10^{-3}$ | $2.87\times 10^{-4}$ |
> > Ratio $\|E(\Delta W_i \Delta  W_j) / E(\Delta W_i^2)\|$ | 6 | $3.52\times 10^{-4}$ | $8.63\times 10^{-5}$ | $1.14\times 10^{-3}$ |  $7.71\times 10^{-5}$ |
> > On-diagonal $E(\Delta W_i^2)$ | 4 | $7.78 \times 10^{0}$ | $6.51 \times 10^{0}$ | $5.71\times 10^{0}$ | $23.09 \times 10^{0}$ |
> > Off-diagonal $E(\Delta W_i \Delta  W_j)$ | 4 | $5.70 \times 10^{-4}$ | $1.13 \times 10^{-3}$ | $1.1\times 10^{-3}$ | $2.43 \times 10^{-3}$|
> > Ratio $\|E(\Delta W_i \Delta  W_j) / E(\Delta W_i^2)\|$ | 4 | $7.43 \times 10^{-5}$| $1.76 \times 10^{-4}$|$1.89 \times 10^{-4}$|$1.08 \times 10^{-4}$|
> >
> >
> >
> > Thanks again for all your very valuable comments. We hope our responses clarify your concerns and questions. Please also let us know if you have further concerns, and we will be glad to continue to address them.

---

### Official Review · Reviewer_VhYT · 2025-10-30

**Soundness:** 2
**Presentation:** 3
**Contribution:** 2
**Rating:** 4
**Confidence:** 4

**Summary:**

This paper proposes a Mixed-Precision Post Training Quantization (PTQ) method for Vision Transformers (ViTs) that minimizes model distortion (the final layer's output error) instead of layer-wise distortion to better preserve accuracy. By approximating model distortion using a second-order Taylor expansion and exploiting its additivity, a dynamic programming algorithm efficiently finds the optimal bit allocation, achieving 4–6 bit quantization without accuracy loss across multiple ViT models.

**Strengths:**

1. The writing and theoretical derivation of this work are reasonable and easy to follow.
2. The problem formulation and solution both take computational efficiency into account.
3. Experiments demonstrate the effectiveness of the proposed method.

**Weaknesses:**

1. The proposed method is not entirely new, similar ideas have been explored in prior works. For example, model output reconstruction instead of layer-wise reconstruction has already been proposed; the second-order optimization in Section 3.1 and the approximation method in Section 4 have also been widely used. If I have misunderstood, the authors are welcome to clarify.
2. The compared methods are somewhat outdated, limited to baselines from 2023 or earlier. Moreover, the comparison might be unfair, as most existing methods perform uniform-precision quantization.
3. Some mathematical derivations could be improved. For instance, in Property 1 (Equation 4), a more rigorous expression should be
E <= E_W + E_A. This can be easily proven and makes more sense than the current “Second-Order Additivity” assumption. It would not affect the subsequent derivation or conclusions, as it can be regarded as optimizing the upper bound of the error.

**Questions:**

0. Please first address the questions in the “weaknesses.”
1. It would be helpful to further summarize the technical contributions. At present, most components appear to have been proposed before, so it would be good to highlight what is newly introduced or improved.
2. When solving the mixed-precision quantization problem, could the authors elaborate on the advantages of using dynamic programming? More commonly used alternatives today maybe are Pareto or linear programming methods.
3. The concept of Model-level Optimization is unclear. For example, although BLOB-Q formulates a global optimization problem, it also decomposes it into subproblems for solving. Similarly, HAWQ models the Hessian of the task loss with respect to model outputs, so why is that not considered “Model-level Optimization”?

Minor comments:
1. In Related Works, the subheadings “PTQ” and “MPQ” do not need to be abbreviated again since they have already been defined earlier.
2. In Section 3, there is only one subsection (3.1). Consider adding an additional subsection for better structure.

---

> ### Author Response · Authors · 2025-11-21
> **Response to Weakness 1**
>
> > ### (W1) The proposed method is not entirely new, similar ideas have been explored in prior works. For example, model output reconstruction instead of layer-wise reconstruction has already been proposed; the second-order optimization in Section 3.1 and the approximation method in Section 4 have also been widely used. If I have misunderstood, the authors are welcome to clarify.
>
>  Thanks to your very good question. We also noticed that there are indeed some prior works that are more or less using similar components with ours, such as model-level optimization, second-order approximation, and Hessian approximation. To our best knowledge, these closely related prior PTQ methods include HAWQ[1], OBQ [2], BRECQ [3], PTQ4ViT [4], LAPQ [5] and HAQ [6]. Next, we clarify the uniqueness of our method against each of them in detail.
>
> - **Model-output reconstruction**: Although these works such as HAQ, HAWQ and LAPQ can be also considered optimizing the model-output, they are different with our method in several aspects: (1) they may use the model-output to optimize other objective (e.g. quantization stepsizes) instead of mixed-precision bit-widths; (2) they develop less efficient optimization methods for the model output; (3) they all optimize on CNN architectures instead of ViTs. We are the first who utilize model-output optimization on ViT mixed-precision quantization, and provide a global-optimal solution within polynomial time. Specifically, HAQ uses global loss to guide reinforcement-learning search which is time consuming; HAWQ similarly uses global loss, but it linearly decides bit-widths which is local optimal; LAPQ uses similar global loss but utilizes a Powell’s algorithm with quadratic-complexity but was proposed only for searching quantizer stepsizes $\Delta$ instead of mixed-precision bit-widths.
> - **Second-order approximation**: As for the taylor second-order approximation that we introduced in Section 3.1, prior arts including OBQ, BRECQ, PTQ4ViT and LAPQ also adopt similar second-order approximation, but they apply it for different problem settings. We aim to leverage the most suitable approximation scheme specific for the unique problem setting in this paper, and perform novel adaptation of existing formations to best suit our case, which is to jointly optimize the mixed-precision bit-widths for both weights and activations. Specifically, OBQ and BRECQ leverage second-order approximation on global loss but with the purpose of learning best quantization rounding parameters. LAPQ and PTQ4ViT use Hessian information to guide layerwise quantizer stepsizes $\Delta$ and are under uniform quantization setting. Furthermore, OBQ, BRECQ and PTQ4ViT all follow a similar trick that transform the computational expensive Hessian of weight $H^w$ to the Hessian on the layerwise output $H^z$ in their final formulation, which is not possible for us since we simultaneously quantize weights and activation.
> - **Hessian approximation**: We are the first to propose an effective Hessian Approximation scheme for both weights and activations on ViTs, while all the previous methods only work on weights. Prior arts adopt different schemes to approximate Hessian for their unique problem settings, such as Gaussian curvature adopted by LAPQ, FIM adopted by OBQ, BRECQ and PTQ4ViT Hessian eigenvalue adopted by HAWQ. However, we adopt empirical Fisher to approximate Hessian on ViTs, which was originally used for pruning, to effectively unify Hessian approximation into a joint search space for both weights and activations. Furthermore, we recognize and address the scalability challenge of the original empirical Fisher on ViTs and derive a scalable empirical Fisher formation (Eq.8 and Eq.9).
>
> Hope it addresses your concerns, and please let us know if any other remaining ones. Thanks again for your comments.

---

> ### Author Response · Authors · 2025-11-21
> **Response to Weakness 2**
>
> > ### (W2) The compared methods are somewhat outdated, limited to baselines from 2023 or earlier. Moreover, the comparison might be unfair, as most existing methods perform uniform-precision quantization.
>
> Thank you for the good suggestions. During rebuttal, we have included multiple more recent mixed-precision approaches in the comparison. Then to have a fairer comparison, we also integrate various uniform-quantization baseline approaches into with a recent SOTA mixed-precision method, then compared their mixed-precision results with us.
>
> -   **Complete Comparisons**:
>
> Specifically, we compare with 3 more recent mixed-precision methods on ViTs, including PMQ [7] (IJCNN' 23), P$^2$-ViT [8] (VLSI’ 24), and PTMQ [9] (AAAI' 24) as shown in Table R1.1. We can observe that our proposed method maintains it advantages with the newer methods coming in. Our method outperforms other methods by 3.08%, 4.45% and 13.4% for 4/6/8-bit respectively on ViT-S. We observe similar improvements on other three models. We hope this address your potential concern on the lack in comparisons, and please let us know if any other baselines are missing.
>
> -   **Fairer Comparison with Uniform-precision Baselines**:
>
> During the rebuttal period, we integrated a SOTA mixed-precision method into various uniform-precision baselines to enable a fairer comparison. Specifically, we choose a latest mixed-precision method P$^2$-ViT to perform integration. We conduct the integration for as many as uniform-precision PTQ baselines with published code available, i.e., RepQ-ViT, EasyQuant, NoisyQuant and PTQ4ViT. We integrate each of their quantizer into P$^2$-ViT evolutionary bit searching process to generate corresponding mixed-precision results.
>
> We show the full comparison under mixed-precision setting in Table R1.2. Firstly, we compare the mixed-precision setting and uniform-precision setting for 4 uniform-precision methods mentioned above. We notice that the mixed-precision results of these uniform-precision methods still underperform our method. Under 4-bit, our method outperforms RepQ-ViT, EasyQuant, NoisyQuant and PTQ4ViT by at least 2.67%, 8.65%, 7.44% and 18.01% respectively. Under 8-bit and 6-bit where bit-widths are high enough, we also notice similar improvement.
>
> This verifies that the superior results of our method indeed come from the proposed mixed-precision approach rather than adopting mixed-precision scheme itself. We have included the above discussions in the revised manuscript. Thanks again for this constructive question.
>
> Table R1.1 Comparison with more recent Mixed-precision methods.
> | Method | Wbit | Abit | ViT-S | ViT-B | DeiT-S | DeiT-B |
> |---|--|---|--|---|--|--|
> | P$^2$-ViT | 4 MP | 8 | 64.24 | 79.93 | 75.26 | 79.37 |
> | BLOB-Q (Ours) | 4 MP | 4 MP | **67.32** | **79.97** | **77.42** | **79.48** |
> | PMQ [7] | 6 MP | 6 MP | - | 73.33 | 76.68 | 79.64 |
> | P$^2$-ViT [8] | 6 MP | 8 | 32.05 | 82.1 | 77.56 | 80.59 |
> | PTMQ [9] | 6 MP | 6 MP | 76.09 | 77.7 | 78.74 | 80.81 |
> | Ours | 6 MP | 6 MP | **80.53** | **83.66** | **79.53** | **81.58** |
> | P$^2$-ViT | 8 MP | 8 MP |68.14 | 83.00 | 78.41 | 80.93 |
> | Ours MP | 8 MP |8 MP | **81.54** | **84.89** | **81.25** | **81.75** |
>
> Table R1.2 Comparisons under mixed-precision settings.
> | Method | Bit | ViT-S | ViT-B | DeiT-S | DeiT-B |
> |--|--|--|---|---|--|
> | Full Precision | 32 | 81.39 | 84.54 | 79.87 | 81.85 |
> | RepQ-ViT | 4 | 64.43 | 66.27 | 67.67 | 75.80 |
> | RepQ-ViT + P$^2$-ViT | 4 MP | 64.65 | 67.08 | 68.15 | 79.18 |
> | EasyQuant | 4 | 34.94 | 71.03 | 28.15 | 68.85 |
> | EasyQuant + P$^2$-ViT | 4 MP | 37.21 | 71.32 | 30.08 | 69.13 |
> | NoisyQuant | 4 | 50.36 | 69.38 | 38.46 | 72.84 |
> | NoisyQuant + P$^2$-ViT | 4 MP | 51.35 | 69.73 | 38.63 | 72.86 |
> | PTQ4ViT | 4 | 40.73 | 55.33 | 27.01 | 61.42 |
> | PTQ4ViT + P$^2$-ViT | 4 MP | 40.81 | 60.73 | 29.64 | 62.29 |
> | Ours MP | 4 MP | **67.32** | **79.97** | **77.42** | **80.30** |
> | RepQ-ViT | 6 | 80.14 | 83.39 | 78.69 | 81.16 |
> | RepQ-ViT + P$^2$-ViT | 6 MP | 80.15 | 83.41 | 78.81 | 81.23 |
> | EasyQuant | 6 | 77.27 | 81.54 | 65.08 | 79.34 |
> | EasyQuant + P$^2$-ViT | 6 MP | 77.38 | 82.85 | 67.93 | 79.55 |
> | NoisyQuant | 6 | 77.72 | 82.10 | 65.12 | 79.67 |
> | NoisyQuant + P$^2$-ViT | 6 MP | 77.75 | 82.48 | 67.59 | 79.74 |
> | PTQ4ViT | 6 | 78.02 | 82.90 | 76.29 | 80.19 |
> | PTQ4ViT + P$^2$-ViT | 6 MP | 78.53 | 83.09 | 76.37 | 80.29 |
> | Ours MP | 6 MP | **80.73** | **84.23** | **79.53** | **81.85** |
> | RepQ-ViT | 8 | 81.23 | 84.43 | 78.69 | 81.16 |
> | RepQ-ViT + P$^2$-ViT | 8 MP | 81.23 | 84.49 | 79.69 | 81.74 |
> | EasyQuant | 8 | 80.87 | 84.77 | 78.79 | 81.34 |
> | EasyQuant + P$^2$-ViT | 8 MP | 80.96 | 84.80 | 78.87 | 81.37 |
> | NoisyQuant | 8 | 80.92 | 84.76 | 78.89 | 81.32 |
> | NoisyQuant + P$^2$-ViT | 8 MP | 81.02 | 84.84 | 79.05 | 81.40 |
> | PTQ4ViT | 8 | 80.90 | 84.63 | 79.43 | 81.48 |
> | PTQ4ViT + P$^2$-ViT | 8 MP | 80.93 | 84.70 | 79.53 | 81.54 |
> | Ours MP | 8 MP | **81.54** | **84.89** | **81.25** | **81.75** |

---

> ### Author Response · Authors · 2025-11-21
>
> > ### (W3) Some mathematical derivations could be improved. For instance, in Property 1 (Equation 4), a more rigorous expression should be E <= E_W + E_A. This can be easily proven and makes more sense than the current “Second-Order Additivity” assumption. It would not affect the subsequent derivation or conclusions, as it can be regarded as optimizing the upper bound of the error.
>
> Thanks for the very good insight and suggestion. We strongly agree with your suggestion and we have conducted further analysis and derivation on the inequality version of Property 1. We indeed obtained the conclusion. We will adopt your suggestion and amend the discussion of this inequality in the manuscript, as we explain in detail below.
>
> Specifically, we found that the expectation of weight and activation quantization errors on model output indeed follow an inequality. We first perform Taylor second-order approximation on the model output distortion considering all layer quantization:
> $$\Gamma(\mathbf{O},\hat{\mathbf{O}})\approx \mathbf{E}(||\sum_l \frac{1}{2}\Delta\mathbf{W}_l^\top \mathbf{H}_l^w \Delta\mathbf{W}_l +\sum_l\frac{1}{2} \Delta\mathbf{A}_l^\top \mathbf{H}_l^a \Delta\mathbf{A}_l||).$$
> Then we can apply triangular inequality property of the L1-norms of two variables $\sum_l \frac{1}{2} \Delta\mathbf{W}_l^\top \mathbf{H}_l^w \Delta\mathbf{W}_l$ and $\sum_l \frac{1}{2}\Delta\mathbf{A}_l^\top \mathbf{H}_l^a \Delta\mathbf{A}_l$, giving
> $$||\sum_l \frac{1}{2}\Delta\mathbf{W}_l^\top \mathbf{H}_l^w \Delta\mathbf{W}_l +  \sum_l \frac{1}{2}\Delta\mathbf{A}_l^\top \mathbf{H}_l^a \Delta\mathbf{A}_l|| \leq ||\sum_l \frac{1}{2}\Delta\mathbf{W}_l^\top \mathbf{H}_l^w \Delta\mathbf{W}_l|| + ||\sum_l \frac{1}{2}\Delta\mathbf{A}_l^\top \mathbf{H}_l^a \Delta\mathbf{A}_l||.$$
> Therefore we have $\Gamma(\mathbf{O}, \hat{\mathbf{O}}) \leq \mathbf{E}(||\sum_l \frac{1}{2}\Delta\mathbf{W}_l^\top \mathbf{H}_l^w \Delta\mathbf{W}_l||) + \mathbf{E}(||\sum_l \frac{1}{2}\Delta\mathbf{A}_l^\top \mathbf{H}_l^a \Delta\mathbf{A}_l||)$. Since summations inside expectation can be swapped out of expectation, we have
> $$\Gamma(\mathbf{O}, \hat{\mathbf{O}}) \leq \sum_l\mathbf{E}(|| \frac{1}{2}\Delta\mathbf{W}_l^\top \mathbf{H}_l^w \Delta\mathbf{W}_l||) + \sum_l\mathbf{E}(|| \frac{1}{2}\Delta\mathbf{A}_l^\top \mathbf{H}_l^a \Delta\mathbf{A}_l||),$$
>
> that is
> $$\Gamma(\mathbf{O}, \hat{\mathbf{O}}) \leq \sum\_l\Gamma_{\hat{W}\_l}(\mathbf{O}, \hat{\mathbf{O}}) + \sum\_l\Gamma\_{\hat{A}\_l}(\mathbf{O}, \hat{\mathbf{O}}).$$
>
> We indeed agree that interpreting the additivity property in the form of upper bound provides an alternative explanation which is mathematically reasonable. Thank you again for this valuable insight.
>
> > ### (Q2) It would be helpful to further summarize the technical contributions. At present, most components appear to have been proposed before, so it would be good to highlight what is newly introduced or improved.
>
> We summarize our innovative and improved components in this paper as below:
> -   **Model-level optimization**: We formulate a model-output optimization for mixed-precision quantization for ViTs, and we include both weights and activations in the problem domain. Although prior arts leverage model-output optimization, such as LAPQ, OBQ, BRECQ and PTQ4ViT, they only target on different problems under uniform quantization like finding the stepsizes and rounding directions, instead of bit-widths in mixed-precision quantization. Other mixed-precision methods, such as HAQ and HAWQ, only consider bit-allocations for weights. In contrast, jointly optimizing bit-widths for weights and activations has larger flexibility in reducing the accuracy drops.
> - **Additivity Approximation and Hessian approximation**: We novelly proposed the important additivity property in mixed-precision quantization, which enables solving the optimization problem efficiently. As for the Hessian approximation, we are the first to leverage Hessian approximation simultaneously for weight and activation in a unified formulation, and we adopt empirical Fisher scheme to compute Hessians. To address the scalability challenge of the original empirical Fisher on ViTs, we improve the original formulation and derive a scalable formation of empirical Fisher.
> - **Optimization Solution**: The proposed dynamic-programming solver achieves the optimality of the solution and the efficiency of the solution. We are able to obtain global-optimal solution for mixed-precision bit-widths within quadratic time. Existing mixed-precision methods are not able to achieve tractable global-optimality.
>
> As the result, the proposed method achieved remarkable low-bit results on ViTs, where we significantly improve the SOTA results on 4-bit post-training quantization, by at least 1.72%, 11.49%, 6.15% and 3.87% for ViT-S, ViT-B, DeiT-S, DeiT-B respectively. Hope this addresses your concerns and improves the clarity on our contributions, and please let us know if any unclarity remains.

---

> ### Author Response · Authors · 2025-11-21
>
> > ### (Q3) When solving the mixed-precision quantization problem, could the authors elaborate on the advantages of using dynamic programming? More commonly used alternatives today maybe are Pareto or linear programming methods.
>
> Many thanks for pointing out the alternative optimization methods. Compared to pareto or linear programming, the main advantage of dynamic programming is that it guarantees global optimal solution within quadratic time complexity. Note that the bit-width allocation problem for model is an integer programming problem, because the optimization variables which are the layerwise bit-widths are discrete integer numbers. Therefore it is more natural to consider it as a discrete optimization problem. Both Pareto or linear programming guarantees only local optimality under discrete problem domain instead of global optimality in dynamic programming.
>
> In addition, during rebuttal period, we have conducted additional studies to verify the superiority of the dynamic programming. Specifically, we replace the dynamic programming algorithm with Pareto or linear programming, and then compare the results. We implement the Lagrangian algorithm to linearly find the pareto frontier given a Lagrangian multiplier, which determines the slope of the linear approacher.
>
> As the Table below shows, we compare the performance of Pareto method to our dynamic programming-based method under 4/6/8-bits. We observe that the proposed dynamic programming solution consistently outperforms linear Pareto method by up to 62.5%, 11.78% and 4.54% for 4/6/8-bit respectively. This evaluates that the proposed dynamic programming solver indeed finds better solution than local-optimal algorithms.
>
> Table R1.3 Dynamic Programming v.s. Pareto Method
> | Model | Bit| Pareto Method | Dynamic Programming (Ours) |
> |---|---|---|---|
> | ViT-S (81.39%) | 4 | 5.07 | **67.32** |
> |  | 6 | 68.75 | **80.53** |
> |  | 8 | 79.68 | **81.54** |
> | ViT-B (84.54%) | 4 | 72.58 | **79.97** |
> |  | 6 | 82.81 | **83.66** |
> |  | 8 | 80.3 | **84.84** |
> | ViT-B/384 (86.05%) | 4 | 57.81 | **71.2** |
> |  | 6 | 80.14 | **85.2** |
> |  | 8 | 85.3 | **85.93** |
> | DeiT-S (79.87%) | 4 | 64.39 | **75.18** |
> |  | 6 | 78.56 | **79.22** |
> |  | 8 | 79.73 | **81.25** |
> | DeiT-B (81.85%) | 4 | 73.56 | **79.48** |
> |  | 6 | 79.55 | **81.58** |
> |  | 8 | 80.21 | **81.75** |
> | DeiT-B/384 (83.12%) | 4 | 72.65 | **80.17** |
> |  | 6 | 81.25 | **82.72** |
> |  | 8 | 81.71 | **82.85** |
>
> > ### (Q4) The concept of Model-level Optimization is unclear. For example, although BLOB-Q formulates a global optimization problem, it also decomposes it into subproblems for solving. Similarly, HAWQ models the Hessian of the task loss with respect to model outputs, so why is that not considered “Model-level Optimization”?
>
> Thank you for the great question. Model-level optimization means to formulate the optimization objective using model-level information and solve this objective jointly considering all the layers or components in the model. We realize that there is a typo in the description for HAWQ as well as in Fig.1. The HAWQ shall be considered as under model-level optimization, as they leverage global task loss to guide the layerwise bit-allocation, and the fact that the mixed-precision bit-widths are determined by global ranking jointly with all layers. We will fix the corresponding introduction of HAWQ in the related sections in the revised manuscript. Thank you again for the very careful review.
>
> > ### (MC1) In Related Works, the subheadings “PTQ” and “MPQ” do not need to be abbreviated again since they have already been defined earlier.
> Thanks for the suggestion. We have removed the redundant abbreviations in the related work section in the revised version.
>
> > ### (MC2) In Section 3, there is only one subsection (3.1). Consider adding an additional subsection for better structure.
> Thanks again for the very good suggestion on improving our manuscript. Following this suggestion, we have modified the structure of Section 3 into 3 subsections, including Layer Distortion Minimization, Model Distortion Minimization and Second-order Approximation and Additivity Property.
>
> **References**
>
> [1] "Hawq: Hessian aware quantization of neural networks with mixed-precision." ICCV'19.
>
> [2] "Optimal brain compression: A framework for accurate post-training quantization and pruning."  NeurIPS'22.
>
> [3] "BRECQ: Pushing the Limit of Post-Training Quantization by Block Reconstruction." ICLR'21.
>
> [4] "Ptq4vit: Post-training quantization for vision transformers with twin uniform quantization." ECCV'22.
>
> [5] "Loss aware post-training quantization". ML'21.
>
> [6] "Haq: Hardware-aware automated quantization with mixed precision." CVPR'19.
>
> [7] Patch-wise mixed-precision quantization of vision transformer. IJCNN '23.
>
> [8] P^2-ViT: Power-of-Two Post-Training Quantization and Acceleration for Fully Quantized Vision Transformer. VLSI' 24.
>
> [9] Post-training multi-bit quantization of neural networks. AAAI' 24.

---

> > ### Comment · Reviewer_VhYT · 2025-11-26
> >
> > Thank you for your rebuttal, but I still have significant concerns regarding the contributions of the paper.
> >
> > 1. In W1's response,
> >
> > (1) “they may use the model-output to optimize other objectives (e.g., quantization step sizes) instead of mixed-precision bit-widths” is incorrect. Both HAQ and HAWQ are applied to mixed-precision bit-widths.
> >
> > (2) “They all optimize on CNN architectures instead of ViTs.” How does the authors' proposed method demonstrate adaptability to ViTs? Which specific components of ViTs does it optimize?
> >
> > (3) “We are the first to propose an effective Hessian Approximation scheme for both weights and activations on ViTs” is incorrect. HAWQ has already discussed this, as has QDrop. Additionally, there are works discussing this in the quantization of diffusion models, where activation distributions are even more challenging to handle.
> >
> > 2.  In W2's response, the authors incorporated newer mixed-precision quantization methods. However, I believe a more compelling approach would be to include additional recent PTQ methods and combine these PTQ methods with classical mixed-precision quantization techniques. Currently, the PTQ methods cited are limited to RepQ-ViT, EasyQuant, NoisyQuant, and PTQ4ViT, all of which are over two years old.

---

> > > ### Author Response · Authors · 2025-12-02
> > >
> > > Thanks for your careful review and follow-up comments. We address your follow-up comments one-by-one as below.
> > >
> > > > ### (1) “they may use the model-output to optimize other objectives (e.g., quantization step sizes) instead of mixed-precision bit-widths” is incorrect. Both HAQ and HAWQ are applied to mixed-precision bit-widths.
> > >
> > > Sorry for the confusion. By ‘they’ we referred to methods like LAPQ. For HAQ and HAWQ, we classify them as mixed-precision methods, and they are different to our method in other aspects. We have discussed their differences with us in detail. Specifically, to perform mixed-precision bit-allocation, HAQ uses global loss to guide Reinforcement Learning search, and HAWQ uses linear optimization. Both are less optimal solutions compared to us. In this work, we proposed an important additivity property which effectively reformulates the original NP-hard problem of minimizing model output distortion into a feasible form. Such reformulated objective comes with tractable solutions within polynomial time. We further proposed a highly efficient optimization algorithm that finds global optimal solution with quadratic time complexity. Such global optimal solution leads to significant model accuracy improvements.
> > >
> > > > ### (2) “They all optimize on CNN architectures instead of ViTs.” How does the authors' proposed method demonstrate adaptability to ViTs? Which specific components of ViTs does it optimize?
> > >
> > > By "optimize" we meant that those baselines were proposed focusing on CNN models, and did not evaluate their performances on ViT models. Our method was systematically evaluated on ViT models. We apologize if the relevant statement causes any misunderstanding, and we would avoid such expression.
> > >
> > > Our method demonstrates significant advantages on ViTs thanks to the efficiency and scalability of our solution on largely parameterized and computationally complex models. Compared to CNN models, ViTs contain much more parameters and computation overhead. Facing this scalability challenge of ViT models, we designed a scalable empirical Fisher Hessian approximation scheme that calibrates ViTs with linear time complexity, and we proposed a highly efficient solution with quadratic time complexity.
> > >
> > > As the proposed method is highly efficient with high accuracy, it is particularly suitable for models with high computational complexity like ViTs, and can obtain superior performance even on low bits.
> > >
> > > > ### (3) “We are the first to propose an effective Hessian Approximation scheme for both weights and activations on ViTs” is incorrect. HAWQ has already discussed this, as has QDrop....
> > >
> > > Regarding the Hessian Approximation scheme jointly on weights and activations, we wish to highlight that existing methods are noticeably different from us, including HAWQ and QDrop. To our best knowledge, although existing methods may have considered optimizing the activation quantization, they only consider the impact of either weights or activations separately in their quantization objectives, instead of modelling both weights and activations in a joint objective. We take HAWQ and QDrop as examples, and we explain their differences to us in detail as below.
> > >
> > > For HAWQ, although they discussed Hessian information for bit-allocation, they did not formulate the impact of activation bit-widths towards their objective function, but instead they first search for bit-widths for weights of a layer/block, and then apply the same bit assignment of weights directly to the activations. Furthermore, the hessian-based proxy they derived for bit-allocation only formulates on weights rather than on activations. In contrast, our work explicitly formulates the impacts of both weights and activations on quantization separately.
> > >
> > > For QDrop, although they indeed incorporate activation quantization during the diffusion models calibration, their method actually does not use Hessian to optimize their quantization objective, and they only discuss Hessian to provide lateral support for the smoothing effect of their method in the additional experimental discussion. Moreover, they focus on the CNN-based diffusion models.
> > >
> > > Finally, to better illustrate the contributions of this work, we outline the key differences and contributions below:
> > >
> > > 1. **Objective with Additivity Property**: We proposed an important additivity property on the output distortion minimization objective. The original objective is an NP-hard problem. With the proposed property, the objective is much easier to solve.
> > > 3. **Efficient and Optimal Solution**: We proposed an algorithm that is able to obtain global optimal solution within polynomial time. The global optimal solution leads to a high accuracy.
> > > 4. **Noticeable Result Improvements over Prior Arts**: Our approach noticeably improves state-of-the-art on ViTS, especially under low bit-widths.

---

> > > > ### Author Response · Authors · 2025-12-02
> > > >
> > > > > ### 3.  In W2's response, the authors incorporated newer mixed-precision quantization methods. However, I believe a more compelling approach would be to include additional recent PTQ methods and combine these PTQ methods with classical mixed-precision quantization techniques. Currently, the PTQ methods cited are limited to RepQ-ViT, EasyQuant, NoisyQuant, and PTQ4ViT, all of which are over two years old.
> > > >
> > > > Thanks for the constructive comment. Following your suggestion, we have combined 2 more recent uniform-precision PTQ methods, ERQ (ICML'24) [10] and SARDFQ (ICCV'25) [11], with classical mixed-precision quantization techniques, and then compare the performance with our method. Specifically, we choose P$^2$-ViT as the mixed-precision bit allocation technique to combine with the quantizers of the two PTQ methods respectively. Table R1.4 illustrates the results in detail. In Table R1.4, we observe similar behaviors as on the previous integration studies, where the mixed-precision results of ERQ and SARDFQ still underperform our method. Under 4-bit, our method outperforms ERQ and SARDFQ by at least 0.35% and 6.71% respectively. Under 6-bit, we also notice similar improvement. This shows that our method still has advantage against recent PTQ methods under mixed-precision bit-widths.
> > > >
> > > >
> > > >  Table R1.4 Integrating additional PTQ methods with existing mixed-precision technique.
> > > > | Method | Bit | ViT-S | ViT-B | DeiT-S | DeiT-B |
> > > > |-------|------|-------|-------|--------|--------|
> > > > | Full Precision | 32 | 81.39 | 84.54 | 79.87 | 81.85 |
> > > > | ERQ + P$^2$-VIT | 4 MP | 68.97 | 77.06 | 73.15 | 78.56 |
> > > > | SARDFQ + P$^2$-VIT | 4 MP | 51.89 | 53.41 | 63.42 | 73.59 |
> > > > | Ours | 4 MP | 69.32 | 79.97 | 77.42 | 80.3 |
> > > > | ERQ + P$^2$-VIT |6 MP | 80.69 | 84.05 | 79.51 | 81.84 |
> > > > | SARDFQ + P$^2$-VIT | 6 MP | 78.72 | 79.39 | 77.68 | 80.27 |
> > > > | Ours | 6 MP | 80.73 | 84.23 | 79.53 | 81.85 |
> > > >
> > > >
> > > >
> > > > **References**:
> > > >
> > > > [10] "Erq: Error reduction for post-training quantization of vision transformers." ICML'24.
> > > >
> > > > [11] "Semantic Alignment and Reinforcement for Data-Free Quantization of Vision Transformers." ICCV'25.

---

> ### Author Response · Authors · 2025-11-24
> **Please kindly check our response and revision**
>
> Dear Reviewer VhYT,
>
> \
> Thank you again for your careful and extremely insightful comments and many constructive suggestions.
>
> \
> In our previous responses, we have carefully provided detailed explanations and additional analysis to your comments one-by-one. We have also accordingly revised the relevant places in the manuscript.
>
> \
> It would be very appreciated if you can take a look, and please let us know if there are any remaining concerns.
>
> \
> Thanks a lot for your time and effort for the review.
>
> \
> Best regards,
>
> Authors

---

### Comment · Area_Chair_BPMN · 2025-11-27
**REVIEWERS SHOULD ENGAGE IN THE DISCUSSION**

Dear reviewers,

Please check the author's reply. Feel free to raise any questions or start a discussion, regardless of whether you will change the score.

Your AC.

---

### Author Response · Authors · 2025-12-02
**Summary of Rebuttal**

Dear AC,

\
We sincerely thank all reviewers for the valuable comments. During rebuttal period, we have carefully addressed all comments from 4 reviewers. All reviewers have provided feedbacks to our responses, and we have accordingly provided further clarification to their remaining concerns. Due to the rebuttal policy change, we didn't get another round of follow-ups from reviewers. Below, we provide an overall summary of the rebuttal:
- R1 (VhYT): The main concern from R1 is the differences of our method with several other baseline methods. Regarding that, we have explicitly highlighted the differences of our method to each baseline method in detail. R1 also conveyed some slightly inaccurate understanding towards those baselines, and we have carefully clarified and explained them. We also conducted all the additional experimental evaluations required by R1.
- R2 (JeSs): We are glad that R2 has positively conveyed that we **addressed most concerns**. The remaining questions from R2 focus on the minor results in the intermediate statistics of our empirical studies. We have accordingly provided additional discussion on them in detail.
- R3 (qybZ): We are pleased that R3 positively conveyed that we addressed all the concerns. R3 also maintains original score (**8, accept**).
- R4 (HeS4): We are glad that R4 **did not show further concerns** to our response, and only requests for one more experiment comparison with baselines. Accordingly, we have provided related results in details.

\
During rebuttal period, we have tried our best and devoted tremendous efforts to address all the concerns. We have explained all the questions, clarified the misunderstandings and added comprehensive results. The manuscript has also been revised accordingly. We will be appreciated if you could review our discussions with reviewers. We are also willing to provide further clarifications and responses if any needed.

\
Sincerely, Authors

---

> ### Author Response · Authors · 2025-12-03
> **[Remark] Key Contributions and Differentiations.**
>
> We wish to thank again the effort of all Reviewers and AC. We are pleased that reviewers collectively found the paper "**well-motivated and well-studied**" (`qybZ`,`JeSs`,`HeS4`) and "**extensively evaluated**" (`VhYT`,`JeSs`), and appreciated the "**theoretical soundness of our framework**" (`VhYT`,`qybZ`,`JeSs`) and "**dedication on computation efficiency**" (`VhYT`,`HeS4`). The rebuttal discussion and extensive new experiments have substantially strengthened our work.
>
> \
> Below, we wish to reiterate the key contributions of the proposed BLOB-Q and highlight the key differences with prior works asked by reviewer (mainly `VhYT`).
>
> ---
>
> ### Key Contributions and Differences with Prior Works:
>
> 1. **Objective with Additivity Property**:
>   - **Advantage #1**: We show that model-level optimization is actually possible to decompose via the proposed Additivity Property. The original objective is an **NP-hard** problem. With the proposed property, the objective is much more **feasible** to solve.
>   - **Advantage #2**: Our decomposed objective by Additivity Property maintains optimality, validated by accuracy on ViTs.
>   - Prior model-level methods such as HAQ (RL), HAWQ and EMQ (greedy ranking), PTQ4ViT and LAPQ (heuristics / linear search) cannot decompose the global objective and therefore rely on **sub-optimal** solutions.
>
> 2.  **Joint Objective for Mixed-Precision Weights + Activations**:
>   - We are the first to formulate a **joint objective** that explicitly models the impact of both weight and activation bit-widths. Table R4.5 (for `HeS4`) confirms the advantage of joint optimization over weight-only Mixed-precision Quantization (MPQ).
>   - HAWQ/HAQ do not optimize activation bit-widths and only copy weight allocation;
>     EMQ/OMPQ/PMQ/PTMQ/P²-ViT **fix uniform bit-widths** for activations;
>     PTQ4ViT/LAPQ discuss activation but still optimize only weight quantization.
>
> 3. **Efficient and Optimal Solution**:
>   - We proposed an algorithm that is able to obtain global optimal solution within **quadratic complexity**. The global optimal solution leads to a high accuracy.
>   - Prior MPQ baselines are either **local-optimal** (greedy-like ranking, e.g. HAWQ, EMQ, PMQ) or **time consuming** (RL or Evolutionary Search, e.g. HAQ, OMPQ, PMTQ).
>
> 4. **Scalable Hessian Approximation**:
> - We managed to achieve **linear complexity** for the expensive Hessian calculation on ViTs, largely improving the original empirical Fisher approximation.
> - We managed to integrate Hessians for weights and activations into a joint objective, via their dedicated formulations for weights and activations respectively.
> - Prior works that discuss Hessian do not consider a joint optimization scenario:
> 	- HAWQ and LAPQ computes Hessian only for weights, and activation sensitivity is not modeled
> 	- BRECQ, OBQ, PTQ4ViT adopt uniform bit-width for activations. They adopt Hessian of activations in formulation but is eventually for optimizing weight quantization.
> 	- QDrop (mentioned by `VhYT`) does not leverage Hessian in the proposed method, but only mentions in the qualitative analysis.
>
> \
> We hope this remark makes it easier to navigate through the relevant discussions and facilitate the assessment of our work.

---

### Note · Authors · 2026-02-05

I have read and agree with the venue's withdrawal policy on behalf of myself and my co-authors.

---

### Meta-Review · Area_Chair_L9jE · 2025-12-31

**Summary:**

The paper proposes "BLOB-Q," a post-training quantization (PTQ) framework specifically for Vision Transformers (ViTs). The core contribution is a mixed-precision bit allocation strategy that minimizes global model distortion (output error of the last layer) rather than local layer-wise distortion. The authors employ a second-order Taylor expansion to approximate this distortion and utilize a dynamic programming algorithm to solve the allocation problem efficiently.

While the problem of ViT quantization is relevant, the novelty of the proposed solution appears limited and incremental. The "global distortion minimization" and "Hessian-based assessment" techniques are well-established in the literature (e.g., HAWQ, OBQ, BRECQ), and their application here to ViTs, while methodologically sound, does not constitute a significant conceptual breakthrough. Furthermore, the benchmarking—despite rebuttal efforts—remains a point of contention. The reliance on older baselines and the specific manner of integrating mixed-precision strategies into comparisons raises questions about the fairness and strength of the empirical results. Finally, the enthusiastic score of 8 from one reviewer appears unsupported by a critical analysis of the prior art, skewing the overall perception of the paper's contribution.

**Reviewer Concerns:**

### **Addressed Concerns**
**Clarification of Technical Definitions**: The authors successfully clarified the definition of "Model-level Optimization" and addressed typos regarding the classification of HAWQ in their literature review.

**Empirical Evidence for Independence Assumptions**: In response to concerns from Reviewers qybZ and HeS4 regarding the theoretical assumption of inter-layer independence, the authors provided statistical evidence showing that the product of quantization errors between layers is significantly smaller than the squared errors of individual layers.

### **Remaining Concerns**
**Novelty and Incremental Contribution:** This remains the most significant hurdle. Reviewer VhYT strongly contended that the method is not entirely new, citing that model output reconstruction and second-order optimization are widely used techniques (e.g., in OBQ, BRECQ, HAWQ). While the authors argue they are the first to apply this jointly to weights and activations for ViTs, this is largely an adaptation of existing frameworks rather than a novel quantization theory. The "additivity property" is essentially a consequence of the Taylor expansion used in prior work, not a new discovery.

**Benchmark Relevance and Fairness**: Reviewer VhYT noted that the baselines were largely outdated (2023 or earlier). Although the authors added newer comparisons (PMQ, $P^2$-ViT), VhYT maintained that the comparison methodology, specifically integrating mixed-precision strategies into older PTQ methods rather than comparing against the strongest native state-of-the-art, was unconvincing.

**Reviewer Scores:**

The review scores reflect a significant divergence in the assessment of the paper's contribution. Three reviewers (VhYT, qybZ, and HeS4) initially assigned a rating of 4 (Marginally below acceptance), citing concerns about novelty, benchmarking against outdated baselines, and questionable theoretical assumptions. While Reviewer qybZ noted post-rebuttal that most concerns were addressed, Reviewer VhYT explicitly maintained significant concerns regarding the paper's contributions and the validity of the comparisons. In contrast, Reviewer JeSs assigned a score of 8 (Accept), finding the output distortion minimization idea effective and the ablation studies extensive, though this high score appears to overlook the lack of novelty relative to established quantization literature raised by the other reviewers.

---

### Decision · Program_Chairs · 2026-01-26

Reject